# Offline Reinforcement Learning with Adaptive Feature Fusion

**Tieru Wang    Kunbao Wu    Guoshun Nan***

Beijing University of Posts and Telecommunications, China

`{wangtieru,wukunbao,nanguo2021}@bupt.edu.cn`

## ABSTRACT

Return-conditioned supervised learning (RCSL) algorithms have demonstrated strong generative capabilities in offline reinforcement learning (RL) by learning action distributions based on both the state and the return. However, existing approaches treat RL as a conditional sequence modeling task, where actions are predicted from historical context and a target return. This leads to a critical flaw: the policy can overfit to the specific, often suboptimal, actions found within those historical contexts. Consequently, even when conditioned on a high target return, the model struggles to synthesize a correspondingly high-quality action sequence, which fundamentally limits its ability to perform effective trajectory stitching and outperform the behavioral policy. To address these limitations, we propose a novel approach, the Q-Augmented Dual-Feature Fusion Decision Transformer (QDFFDT). Our key innovation is a learnable fusion mechanism that explicitly separates and then adaptively combines global, history-aware sequence features with local, immediate Markovian features. This introduces a structural bias that prioritizes single-step dynamics while still leveraging long-term context, improving generalization without the need for extensive hyperparameter tuning. Experimental results on the D4RL benchmark show that QDFFDT outperforms current state-of-the-art methods, demonstrating the power of adaptive feature fusion for robust offline RL. Our code is available at `https://github.com/wangtieru2/QDFFDT`.

## 1 INTRODUCTION

Offline reinforcement learning (RL) is a data-driven paradigm that learns exclusively from a fixed dataset of previously collected experiences (Levine et al., 2020). This paradigm can be used in scenarios where online interaction is impractical, either because data collection is expensive or dangerous (Kiran et al., 2021). However, the inability to interact with the environment brings some challenges, like out-of-distribution (OOD) problems. Previous works on offline RL generally address this problem by compelling the learned policy to be close to the behavior policy (Fujimoto et al., 2019; Kumar et al., 2019; Wu et al., 2019; Wang et al., 2020; Fujimoto & Gu, 2021) or constraining the value function to assign low values to OOD actions

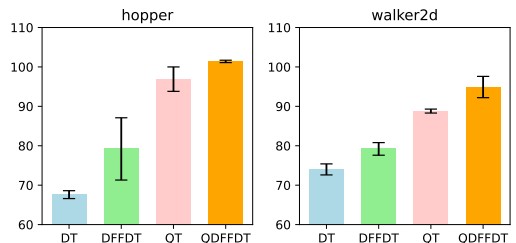

Figure 1: Normalized returns on *medium* datasets. By adaptively fusing two feature streams, our methods leverage suboptimal trajectories and achieve improved performance.

(Kumar et al., 2020; Kostrikov et al., 2021a; Wu et al., 2021). In recent years, with the success of Transformer in CV (Dosovitskiy, 2020; Liu et al., 2021) and NLP (Devlin, 2018; Radford et al., 2019), the architecture has also been introduced to offline RL to model trajectories (Chen et al., 2021; Janner et al., 2021). Based on Transformer, the offline RL problem is viewed as a return-conditioned supervised learning (RCSL) task whose core idea is to learn the return-conditional distribution of

---

*Corresponding author.

actions in each state (Brandfonbrener et al., 2022). By framing RL as a supervised learning problem, RCSL improves both data efficiency and training stability, demonstrating superior performance in offline settings.

However, existing RCSL methods, by treating RL as a pure sequence modeling problem, suffer from a fundamental limitation: they tend to overfit to historical sub-trajectories and thus hinder effective trajectory stitching. A sequence generation model is usually required to reliably reproduce trajectories found within the training dataset, but this objective is inherently misaligned with the goal of RL, which is to discover novel policies that may outperform any single trajectory in the dataset by combining the best segments from multiple different trajectories. Some studies have recognized this issue and employ various approaches to shorten the dependency on long sequences(Wu et al., 2024; Kim et al., 2023; Wang et al., 2025). However, these prior works either lack the flexibility to effectively balance global and local information or require extensive hyperparameter tuning to specific datasets to achieve optimal performance.

Building upon these insights, we propose the Q-Augmented Dual-Feature Fusion Decision Transformer (QDFFDT), a novel architecture that explicitly addresses the tension between global context and local optimality. Our core idea is to learn a state-dependent fusion weight that adaptively balances two distinct feature streams: (1) a **global sequential modeling** stream, which uses a standard self-attention module to capture long-term context, and (2) a **local immediate modeling** stream, which employs a simple MLP to extract features from the current state, respecting the Markov property. This dual-feature design introduces a structural bias that helps the model prioritize high-quality local decisions while not ignoring valuable long-term information, eliminating the need for brittle hyperparameter tuning across different datasets. Furthermore, inspired by (Hu et al., 2024), we augment the RCSL framework with a Q-learning module to enable explicit policy improvement through value-based guidance. As previewed in Figure 1, our method significantly outperforms pure sequence modeling baselines across several suboptimal datasets. Extensive experiments on the D4RL benchmark (Fu et al., 2020) demonstrate that our QDFFDT achieves state-of-the-art performance. Ablation studies further illustrate the flexibility and stability of our method across diverse datasets.

We summarize our contributions as follows: (1) We identify and demonstrate how pure sequence modeling hinders the trajectory stitching ability of the RCSL paradigm. (2) We propose QDFFDT, an adaptive method that integrates global sequential features with local immediate features, providing the necessary inductive bias for offline RL. (3) We demonstrate QDFFDT's superiority over prior methods on D4RL tasks, underscoring its potential to be employed in various scenarios.

## 2    MOTIVATION: WHEN PURE SEQUENCE MODELING HINDERS TRAJECTORY STITCHING

Some prior works (Wu et al., 2024; Kim et al., 2023; Wang et al., 2025) have acknowledged the challenges associated with long-sequence modeling in the context of RL. However, it remains unclear when sequence modeling specifically hinders trajectory stitching in practice. To explain the limitation of sequence modeling, consider a simple task in Figure 2. A sequence modeling method is capable of finding the optimal sub-trajectory "ABCD" in the initial stages. However, after reaching state 'D', the context information can easily mislead the model, causing it to follow a trajectory from the training data towards state 'K', thereby missing the optimal trajectory: "ABCDE". In contrast, a single-step modeling method based on the Markov property is unaffected by historical sub-trajectories and can directly select the action that yields the highest return. We set the sequence length in DT to 1, resulting in the single-step Decision Transformer (SSDT). We then conduct experiments in the simple benchmark environment and measure the success rates of achieving the optimal trajectory "ABCDE". As shown in Table 1, SSDT almost always obtains optimal solutions, whereas DT tends to converge to suboptimal outcomes.

The results of this experiment highlight the inherent difficulty of achieving effective trajectory stitching solely through sequence modeling. Several studies (Yamagata et al., 2023; Wang et al., 2024; Hu et al., 2024) propose augmenting sequence models with value learning to reduce their dependence on historical trajectories. However, whether such hybrid approaches can fully overcome the limitations inherent in sequence modeling remains an open question. To investigate this, we extend our case study by testing the QT (Hu et al., 2024) method. For simplicity, we omit the value learning

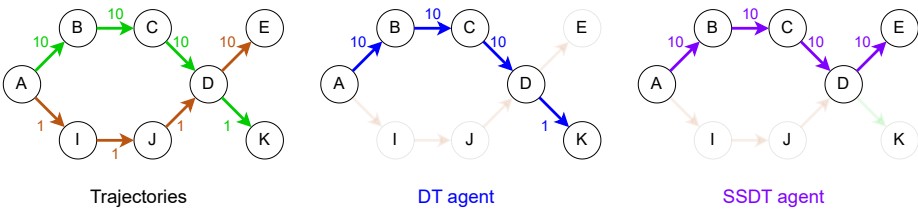

Figure 2: A simple example explains the issue of sequence modeling (DT)—fails to find the trajectory with max return. In contrast, the single-step method (SSDT) finds the optimal path. The numbers on the arrows are rewards and uppercase letters inside the circles represent states. There are two trajectories, "ABCDK" and "AIJDE".

Table 1: The performance of DT, SSDT, and QT with different settings in the simple case. $K$ and $\eta$ are hypeparameters in related methods. Success rates are averaged over 1000 random rollouts (10 independently trained models and 100 trajectories per model).

| Method | $K$ | $\eta$ | Success Rate (%) |
|--------|-----|--------|------------------|
| DT | 3 | - | 0.0 |
| SSDT | 1 | - | 100.0 |
| | 3 | 0.01 | 0.1 |
| | 3 | 0.1 | 0.1 |
| QT | 3 | 1.0 | 0.6 |
| | 3 | 10.0 | 96.2 |

process and instead directly assign action values based on prior knowledge. Reconsidering the example in Figure 2 and setting a discount factor $\gamma = 1.0$, we can compute the Q-values with Dynamic Programming (DP): $Q(D, \nearrow) = 10, Q(D, \searrow) = 1, Q(C, \searrow) = 10 + \max(Q(D, \nearrow), Q(D, \searrow)) = 20$. The values of other actions are computed in a similar manner. Table 1 shows that even if with the introduce of the Q-function to guide policy optimization, the model remains susceptible to the influence of prior sub-trajectories. We hypothesize that this is due to the design of QT's policy objective, which seeks to simultaneously optimize action values while constraining the policy to close to the behavior distribution. As a result, the policy optimization term cannot fully counteract the adverse effects introduced by the behavior cloning term.

## 3  RELATED WORK

### 3.1  OFFLINE REINFORCEMENT LEARNING

Offline RL aims to learn effective policies from static datasets, where a central challenge is mitigating the distributional shift that arises from evaluating policies different from the data-collection policy. A dominant line of work addresses this by constraining the learned policy. For example, CQL (Kumar et al., 2020) learns a conservative, lower-bound Q-function to penalize out-of-distribution actions. Similarly, TD3+BC (Fujimoto & Gu, 2021) incorporates a behavioral cloning loss to regularize the policy, while IQL (Kostrikov et al., 2021b) uses expectile regression to implicitly guide policy improvement. More recently, various generative models have been employed for their capacity to model complex action distributions. These range from A2PO (Qing et al., 2024), which uses conditional VAEs to model advantages, to diffusion models like D-QL (Wang et al., 2022) and DTQL (Chen et al., 2024) that have shown promise in capturing the underlying behavior policy.

### 3.2  RETURN-CONDITIONED SUPERVISED LEARNING

Another line of work tries to solve the offline RL problem by the return-conditioned supervised learning method Brandfonbrener et al. (2022), which aims to learn the return-conditional distribution of actions in each state, and then define a policy by sampling from the distribution of actions that receive high returns. This approach is pioneered by Decision Transformer (DT) (Chen et al.,

2021), which utilizes the sequence model to capture temporal dependencies within entire trajectories. Subsequent works have explored variations on this theme; for instance, DC (Kim et al., 2023) investigate convolution-based backbones, while EDT (Wu et al., 2024) introduce a mechanism to dynamically adjust the model's context length. While powerful, pure return-conditioning can be an unreliable signal for optimality. This limitation has spurred the development of hybrid models that integrate RCSL with explicit value learning. Consequently, ACT (Gao et al., 2024) generates actions conditioned on learned advantages, QT (Hu et al., 2024) combines trajectory modeling with explicit Q-value predictions, and QCS (Kim et al., 2024) adaptively integrates Q-function guidance into the RCSL training objective.

## 4 PRELIMINARIES

### 4.1 OFFLINE REINFORCEMENT LEARNING.

A reinforcement learning problem is typically represented as a Markov decision process (MDP) (Bellman, 1957) $(\rho_0, S, A, P, R, \gamma)$, where $\rho_0$ is the distribution of initial states, $S$ is the state space, $A$ is the action space, $P(s'|s, a)$ is the transition probability, $R(s, a)$ is the reward function, and $\gamma$ is the discount factor. The goal of traditional RL is to train the agent to interact with the environment for a policy $\pi^*(a|s)$ that maximizes the expected return $\mathbb{E}[\sum_{t=0}^{\infty} \gamma^t r(s_t, a_t)]$. Offline RL is different from conventional online RL, we can only train an agent based on the fixed offline dataset $\mathcal{D} = \{(s, a, r, s')\}$ instead of exploring in a simulation environment. The dataset is collected by a behavior policy $\pi_\beta$ from online interaction.

### 4.2 RETURN-CONDITIONED SUPERVISED LEARNING

RCSL methods formulate RL as a supervised learning problem to improve stability. Decision Transformer (DT) (Chen et al., 2021) is one of the important works in RCSL methods. DT models the trajectory as a sequence comprising states, actions, and RTGs. At timestep $t$, DT accepts $\tau = (\hat{R}_{t-K+1}, s_{t-K+1}, a_{t-K+1}, ..., \hat{R}_{t-1}, s_{t-1}, a_{t-1}, \hat{R}_t, s_t)$ as input and outputs $a_t$, where $K$ is the sequence length. The goal of the training stage is to update a neural network so that it converges to the true $a_t$ in the dataset. In the evaluation stage, the real RTG is unavailable. Therefore, DT specifies a target RTG as the desired score. Each timestep, the reward obtained is subtracted from the target RTG until the end of the episode or the maximum length of the trajectory.

While RCSL methods are capable of leveraging historical information for decision-making, their lack of accurate value guidance often hinders their ability to perform effective trajectory stitching. To address this limitation, recent works have proposed to combine value-based learning with sequence modeling for policy optimization (Yamagata et al., 2023; Gao et al., 2024; Hu et al., 2024; Kim et al., 2024).

## 5 METHOD

To address the trajectory stitching challenges inherent in sequence modeling, we consider introducing single-step inputs. However, entirely discarding sequence modeling is impractical, particularly in high-dimensional, pixel-based environments like Atari (Mnih et al., 2013), where compressed state representations often lack the necessary information for optimal decision-making. In such cases, temporal context remains essential. To balance these demands, we propose a hybrid architecture, Dual-Feature Fusion Decision Transformer (DFFDT), which adaptively integrates sequential and immediate contextual features using a learnable weighting mechanism. This design allows the model to dynamically modulate the influence of each feature branch, improving generalization while reducing the burden of manual hyperparameter tuning. Furthermore, to address the known limitation that RTG signals may not reliably reflect action values, we incorporate the Q-learning module, enabling more accurate value estimation and more effective policy improvement. The resulting algorithm is the Q-Augmented Dual-Feature Fusion Decision Transformer (QDFFDT). In this section, we first describe the optimization process for learning the weighting coefficient. We then present a novel RCSL architecture that fuses sequence-level and single-step state features. Finally, we illustrate our value-based policy improvement mechanism.

## 5.1 DUAL-FEATURE FUSION DECISION TRANSFORMER

We investigate how to balance sequential and immediate decision information in RCSL. While sequence modeling excels at reproducing behaviors from offline datasets, it often prioritizes the continuation of previously observed sub-trajectories. This can lead the model to favor actions that are consistent with historical patterns, even if better options exist. On the other hand, single-step modeling makes decisions solely based on RTGs, enabling more flexible trajectory stitching by directly choosing actions associated with higher desired returns. Formally, given a state $\tilde{s}$ with $M$ candidate actions $\{\tilde{a}^i\}_{i=1}^M$ and corresponding RTG values $\{\tilde{R}^i(\tilde{s}, \tilde{a}^i)\}_{i=1}^M$. A single-step model selects $\tilde{a} = \arg\max_{\tilde{a}^i} \tilde{R}^i(\tilde{s}, \tilde{a}^i)$. In contrast, a sequence model, conditioned on preceding sub-trajectories, tends to choose the next action from within a trajectory in the dataset that closely resembles the observed history. To address this issue, we adaptively modulate the influence of sequential features based on whether the training RTGs for $\tilde{s}$ align with the optimal return $\max_i \tilde{R}^i$.

### 5.1.1 LEARNING ADAPTIVE FUSION WEIGHTS

For evaluating the degree of alignment between the training returns and the optimal return, we employ expectile regression (Newey & Powell, 1987; Koenker & Hallock, 2001) to approximate the maximum RTG for a given state. This method is usually used in applied statistics and econometrics. Notably, it has also been integrated into offline RL like IQL (Kostrikov et al., 2021b). We utilize it to estimate the supremum of $\{\tilde{R}^i(\tilde{s}, \tilde{a}^i)\}_{i=1}^M$ for state $\tilde{s}$. The $\sigma \in (0, 1)$ expectile of a random variable $X$ is defined as a solution to the asymmetric least squares problem:

$$\arg\min_{m_\sigma} \mathbb{E}_{x \sim X}[L_2^\sigma(x - m_\sigma)], \tag{1}$$

where $L_2^\sigma(u) = |\sigma - \mathbb{1}(u < 0)|u^2$. When $\sigma > 0.5$, the asymmetric loss assigns more weight to $x$ larger than $m_\sigma$ and less weights to $x$ smaller than $m_\sigma$. Then we can approximate $\max_i \tilde{R}^i$ using a separate value function:

$$\mathcal{L}_V(\psi) = \mathbb{E}_{(s,\hat{R}) \sim \mathcal{D}}[L_2^\sigma(\hat{R} - V_\psi(s))]. \tag{2}$$

The state-value function $V_\psi(s)$ is trained to approximate the empirical upper bound of RTG values in the dataset. Notably, our application of expectile regression differs from that in IQL, which uses it to approximate the optimal Q-value. The purpose of $V_\psi(s)$ is thus to provide a reliable signal for identifying suboptimal data. Specifically, when $V_\psi(s)$ significantly exceeds an observed RTG $\hat{R}(s)$, it indicates the subsequent trajectory is suboptimal. This signal prompts our model to reduce reliance on sequence features, as they might replicate the suboptimal behavior, and instead prioritize Markovian information through an Alpha network:

$$\mathcal{L}_\alpha(\omega) = \mathbb{E}_{(s,\hat{R}) \sim \mathcal{D}}\left[\frac{\alpha_\omega(s) \cdot \max(V_\psi(s) - \hat{R}, 0)}{T}\right], \tag{3}$$

where $T$ is a temperature hyperparameter, and $\alpha_\omega(s) \in (0, 1)$ is produced via a final Sigmoid layer. This final fusion coefficient for sequential features is computed as $\hat{\alpha}(s) = \alpha_{\min} + (1 - \alpha_{\min}) \cdot \alpha_\omega(s)$, where $\alpha_{\min}$ is a predefined minimum sequential weight. Correspondingly, Markovian features are weighted by $1 - \hat{\alpha}(s)$. To encourage strong reliance on sequence modeling at the beginning of training, we initialize the final linear layer of the Alpha network with zero weights and a large positive bias (e.g., 5), ensuring $\alpha_\omega(s) \approx 1$ initially. As training progresses, the loss $\mathcal{L}_\alpha(\omega)$ penalizes $\alpha_\omega(s)$ when $V_\psi(s) > \hat{R}$, driving $\alpha_\omega(s)$ (and thus $\hat{\alpha}(s)$) downwards. Conversely, when $V_\psi(s) \leq \hat{R}$, the loss term vanishes, allowing $\alpha_\omega(s)$ and $\hat{\alpha}(s)$ to remain high, thereby preserving the contribution from sequence modeling for high-quality trajectories.

### 5.1.2 CONTEXTUAL AND MARKOVIAN FEATURE INTEGRATION

Our Dual-Feature Fusion block consists of two key components, as illustrated in Figure 3. Following the design of DD (Ajay et al., 2022), we exclude action tokens from the input sequence, resulting in the following trajectory representation:

$$\tau_t = (\hat{R}_{t-K+1}, s_{t-K+1}, ..., \hat{R}_{t-1}, s_{t-1}, \hat{R}_t, s_t). \tag{4}$$

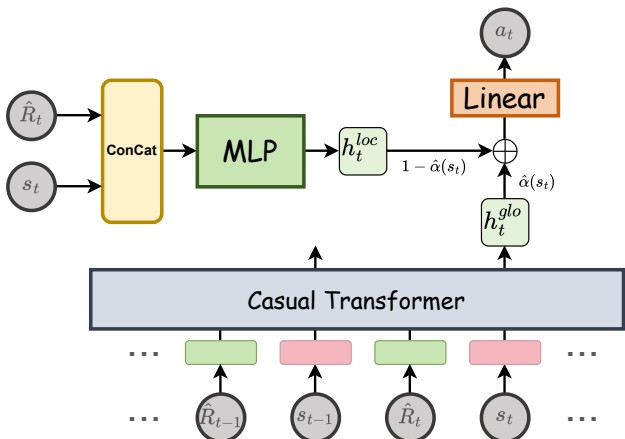

Figure 3: An overview of our Dual-Feature Fusion Decision Transformer architecture. Both states and returns are simultaneously input into the Transformer and MLP components. The resulting intermediate features are adaptively fused via the learned fusion coefficient and then projected into the action space.

Each RTG-state pair $(\hat{R}_t, s_t)$ is encoded into the local representation $h_t^{loc}$ using a lightweight MLP. In parallel, we compute the global representation $h_t^{glo}$ using the self-attention mechanism, as in DT, to capture long-range temporal dependencies. The two feature streams are then fused through the learned coefficient $\hat{\alpha}$ as follows:

$$h = (1 - \hat{\alpha}(s_t)) \cdot h_t^{loc} + \hat{\alpha}(s_t) \cdot h_t^{glo}. \tag{5}$$

The fused representation is finally projected into the action space. To enable effective fusion, both $h_t^{loc}$ and $h_t^{glo}$ are normalized using Layer Normalization (LN) (Ba et al., 2016) to align their feature scales.

During training, consistent with standard RCSL methods, we minimize the Mean Squared Error (MSE) between predicted and ground-truth actions:

$$\mathcal{L}_{DFFDT}(\theta) = \mathbb{E}_{(\tau_t, a_{t-K+1:t}) \sim \mathcal{D}} \left[ \frac{1}{K} \sum_{i=t-K+1}^{t} (a_i - \pi_\theta(\tau_t)_i)^2 \right], \tag{6}$$

where $a_i$ denotes the ground-truth action and $\pi_\theta(\tau_t)_i$ is the $i$-th action predicted by the policy network.

## 5.2 Q-Augmented RCSL Optimization

During training, the RTG often fails to accurately represent the true value of a given state-action pair. Furthermore, the mismatch between the target RTG and the optimal RTG can further degrade the performance of RCSL methods, preventing convergence to the theoretical optimum (Brandfonbrener et al., 2022). Therefore, leveraging the value functions from DP methods to implement policy improvement is necessary. We develop five networks, including two Q-networks $Q_{\phi_1}, Q_{\phi_2}$, two target networks $Q_{\phi'_1}, Q_{\phi'_2}$, and one target policy network $\pi_{\theta'}$. For detailed implementation, we draw inspiration from QT (Hu et al., 2024) and make minor modifications, leading to in the following Q-function loss:

$$\mathbb{E}_{(s_{t-K+1:t+1}, a_{t-K+1:t}, r_{t-K+1:t}) \sim \mathcal{D}} \sum_{m=t-K+1}^{t} \left\| \hat{Q}_m - Q_{\phi_i}(s_m, a_m) \right\|^2,$$

$$\text{where } \hat{Q}_m = \begin{cases} \sum_{j=m}^{t} \gamma^{j-m} r_j + \gamma^{t+1-m} \min_{i=1,2} Q_{\phi'_i}(s_{t+1}, \hat{a}_{t+1}), & \text{(n-step)} \\ r_m + \gamma \min_{i=1,2} Q_{\phi'_i}(s_{m+1}, \hat{a}_{m+1}), & \text{(1-step)} \end{cases} \tag{7}$$

where $\gamma$ is the discount factor. The target actions $\hat{a}_{m+1}$ and $\hat{a}_{t+1}$ are sampled from the target policy $\pi_{\theta'}$. In practice, we manually choose either the n-step or 1-step Bellman equation to estimate the Q-value function. While QT suggests that the n-step Bellman equation demonstrates improvement over the 1-step approximation, we find that this observation does not always hold in experiments. We provide a detailed discussion of this discrepancy in Appendix D.4.

To combine policy improvement with supervised learning, we define the policy loss as a weighted sum of the DFFDT behavior cloning loss and the Q-value guided improvement term:

$$\mathcal{L}_\pi(\theta) = \lambda \cdot \mathcal{L}_{DFFDT}(\theta) + \mathcal{L}_Q(\theta) = \lambda \cdot \mathcal{L}_{DFFDT}(\theta) - \mathbb{E}_{\tau_t \sim \mathcal{D}} \mathbb{E}_{s_i \sim \tau_t} Q_\phi(s_i, \pi_\theta(\tau_t)_i), \quad (8)$$

where $\lambda$ is a weighting coefficient that balances the supervised learning and the value-based policy improvement. Following TD3+BC (Fujimoto & Gu, 2021), the Q-function is normalized to mitigate the scale mismatch across offline datasets and prevent gradient imbalance between the two learning objectives. The detailed pseudo-code is provided by Algorithm 1 in Appendix A.1.

**Convergence Guarantees.** Our approach relies on two well-understood principles: the DFFDT (RCSL) component, whose convergence to the behavior policy has been verified in prior work (Hu et al., 2024), is preserved in our method despite certain architectural adjustments; the Q-augmentation component follows standard dynamic programming, guiding the policy toward higher-value regions with established convergence properties. While a formal proof for the combination is non-trivial, the components are theoretically grounded and validated empirically.

## 6 EXPERIMENTS

Our experiments aim to answer the following questions: (1) How does our QDFFDT perform on the offline benchmarks compared with other state-of-the-art approaches? (2) What advantages does the feature fusion architecture offer compared to modeling with either sequential features or immediate features alone? (3) How does the dynamic fusion of the two components in our QDFFDT differ from static fusion in terms of performance?

**Datasets.** We evaluate our approach on the D4RL benchmark suite (Fu et al., 2020), including tasks from Gym-MuJoCo, Maze2D, AntMaze, Kitchen, and Adroit. Gym-MuJoCo tasks are relatively simple, containing many near-optimal trajectories and smooth reward functions. Maze2D evaluates an agent's ability to stitch sub-trajectories to reach a goal. AntMaze introduces sparse rewards and higher complexity by replacing the 2D ball with an 8-DoF quadruped. Kitchen involves multi-task sequential objectives, emphasizing generalization to unseen states. Adroit is derived from human demonstrations, with a limited state-action space that requires strong policy regularization to ensure stability.

**Baselines.** We compare the proposed QDFFDT with several traditional value-based methods, where we select CQL (Kumar et al., 2020), IQL (Kostrikov et al., 2021b), BEAR Kumar et al. (2019), AWR (Peng et al., 2019), TD3+BC (Fujimoto & Gu, 2021), EQL (Xu et al., 2023), LAPO (Chen et al., 2022), and CQL+AW (Hong et al., 2023). For RCSL and its derivative methods, the evaluated algorithms include DT (Chen et al., 2021), DC (Kim et al., 2023), LSDT (Wang et al., 2025), RvS (Emmons et al., 2021), GDT (Hu et al., 2023), QDT (Yamagata et al., 2023), EDT (Wu et al., 2024), Reinformer (Zhuang et al., 2024), POR (Xu et al., 2022), CGDT (Wang et al., 2024), ACT (Gao et al., 2024), QT (Hu et al., 2024), and QCS (Kim et al., 2024). Additionally, we also compare VAE-based and diffusion-based methods, including BCQ (Fujimoto et al., 2019), A2PO (Qing et al., 2024), Diffuser (Janner et al., 2022), DD (Ajay et al., 2022), and D-QL (Wang et al., 2022). The scores of these baselines are sourced from their respective papers or from our experiments to ensure a fair comparison.

### 6.1 MAIN RESULTS

We evaluate our QDFFDT against a suite of strong baselines, reporting normalized scores on the D4RL benchmark following the standard protocol (Fu et al., 2020). As shown in Table 2, our QDFFDT demonstrates significant performance gains, achieving state-of-the-art or competitive results across all evaluated benchmarks. On suboptimal datasets with Markov property, such as the *medium* and *medium-replay* tasks, our method outperforms QT, which lacks sufficient emphasis on local dependencies for policy training. This advantage extends to the challenging navigation tasks

Table 2: The performance of our QDFFDT and baselines on D4RL tasks. Results for QDFFDT correspond to the mean and standard deviations of normalized scores over 50 random rollouts (5 independently trained models and 10 trajectories per model) for Gym tasks, and over 500 random rollouts (5 independently trained models and 100 trajectories per model) for the other tasks. The dataset names are abbreviated as follows: *medium* to 'm', *replay* to 'r', *expert* to 'e', *umaze* to 'u', *large* to 'l', *diverse* to 'd', and *play* to 'p'. The best average values are marked in bold.

| Gym Tasks | CQL | IQL | TD3+BC | DT | DC | LSDT | ACT | CGDT | QT | QCS | QDFFDT |
|---|---|---|---|---|---|---|---|---|---|---|---|
| halfcheetah-m-e | 91.6 | 86.7 | 90.7 | 86.8 | 93.0 | 93.2 | 96.1 | 93.6 | 96.1 | 93.3 | **97.5 ± 0.6** |
| hopper-m-e | 105.4 | 91.5 | 98.0 | 107.6 | 110.4 | 111.7 | 111.5 | 107.6 | **113.4** | 110.2 | 112.3 ± 0.4 |
| walker2d-m-e | 108.8 | 109.6 | 110.1 | 108.1 | 109.6 | 109.8 | 113.3 | 109.3 | 112.6 | **116.6** | 111.6 ± 0.4 |
| halfcheetah-m | 49.2 | 47.4 | 48.4 | 42.6 | 43.0 | 43.6 | 49.1 | 43.0 | 51.4 | 59.0 | **65.7 ± 1.3** |
| hopper-m | 69.4 | 66.3 | 59.3 | 67.6 | 92.6 | 87.2 | 67.8 | 96.9 | 96.9 | 96.4 | **101.4 ± 0.3** |
| walker2d-m | 83.0 | 78.3 | 83.7 | 74.0 | 79.2 | 81.0 | 80.9 | 79.1 | 88.8 | 88.2 | **94.9 ± 2.7** |
| halfcheetah-m-r | 45.5 | 44.2 | 44.6 | 36.6 | 41.3 | 42.9 | 43.0 | 40.4 | 48.9 | 54.1 | **62.0 ± 1.1** |
| hopper-m-r | 95.0 | 94.7 | 60.9 | 82.7 | 94.2 | 93.9 | 98.4 | 93.4 | 102.0 | 100.4 | **102.7 ± 0.5** |
| walker2d-m-r | 77.2 | 73.9 | 81.8 | 66.6 | 76.6 | 74.7 | 56.1 | 78.1 | 98.5 | 94.1 | **101.0 ± 1.3** |
| **Average** | 80.6 | 77.0 | 75.3 | 74.7 | 82.2 | 82.0 | 79.6 | 82.4 | 89.8 | 90.3 | **94.3** |

| Maze2D Tasks | CQL | IQL | Diffuser | DD | EQL | DT | QDT | EDT | Reinformer | QT | QDFFDT |
|---|---|---|---|---|---|---|---|---|---|---|---|
| maze2d-u | 94.7 | 42.1 | 113.9 | 116.2 | 56.5 | 31.0 | 57.3 | 35.8 | 57.2 | 92.3 | **123.3 ± 19.7** |
| maze2d-m | 41.8 | 34.9 | 121.5 | 122.3 | 36.3 | 8.2 | 13.3 | 18.3 | 85.6 | 156.5 | **157.1 ± 4.0** |
| maze2d-l | 49.6 | 61.7 | 123.0 | 125.9 | 57.0 | 2.3 | 31.0 | 26.8 | 47.4 | **214.5** | 198.9 ± 13.0 |
| **Average** | 62.0 | 46.2 | 119.5 | 121.5 | 49.9 | 13.8 | 33.9 | 28.6 | 63.4 | 154.4 | **159.8** |

| AntMaze Tasks | CQL | IQL | TD3+BC | DT | DC | LSDT | RvS | POR | QT | QCS | QDFFDT |
|---|---|---|---|---|---|---|---|---|---|---|---|
| antmaze-u | 74.0 | 87.5 | 78.6 | 59.2 | 85.0 | 80.0 | 65.4 | 90.6 | 78.3 | 92.5 | **99.8 ± 0.4** |
| antmaze-u-d | **84.0** | 62.2 | 71.4 | 53.0 | 78.5 | 83.2 | 60.9 | 71.3 | 72.7 | 82.5 | 78.4 ± 8.8 |
| antmaze-m-p | 61.2 | 71.2 | 10.6 | 0.0 | 33.2 | 85.5 | 58.1 | 84.6 | 28.7 | 84.8 | **94.8 ± 2.0** |
| antmaze-m-d | 53.7 | 70.0 | 3.0 | 0.0 | 27.5 | 75.8 | 67.3 | 79.2 | 26.7 | 75.2 | **97.0 ± 1.4** |
| antmaze-l-p | 15.8 | 39.6 | 0.2 | 0.0 | 4.8 | - | 32.4 | 58.0 | 1.7 | 70.0 | **79.6 ± 3.3** |
| antmaze-l-d | 14.9 | 47.5 | 0.0 | 0.0 | 12.3 | - | 36.9 | 73.4 | 25.0 | 77.3 | **80.0 ± 3.5** |
| **Average** | 50.6 | 63.0 | 27.3 | 18.7 | 40.2 | - | 53.5 | 76.2 | 38.9 | 80.4 | **88.3** |

| Kitchen Tasks | CQL | IQL | BCQ | BEAR | AWR | D-QL | LAPO | CQL+AW | DT | QT | QDFFDT |
|---|---|---|---|---|---|---|---|---|---|---|---|
| kitchen-partial | 49.8 | 46.3 | 18.9 | 13.1 | 15.4 | 60.5 | 53.7 | 36.0 | 57.9 | **68.1** | 65.9 ± 1.0 |
| kitchen-mixed | 51.0 | 51.0 | 8.1 | 47.2 | 10.6 | 62.6 | 62.4 | 50.5 | 42.5 | 53.0 | **63.9 ± 3.4** |
| **Average** | 50.4 | 48.7 | 13.5 | 30.2 | 13.0 | 61.6 | 58.1 | 43.3 | 50.2 | 60.6 | **64.9** |

| Adroit Tasks | CQL | IQL | D-QL | A2PO | DT | DC | DD | GDT | QT | QCS | QDFFDT |
|---|---|---|---|---|---|---|---|---|---|---|---|
| pen-human | 37.5 | 71.5 | 72.8 | 68.9 | 62.9 | 74.2 | 66.7 | 92.5 | **94.6** | 83.9 | 94.1 ± 7.9 |
| pen-cloned | 39.2 | 37.3 | 57.3 | 85.0 | 28.7 | 50.0 | 42.8 | 86.2 | 85.6 | 66.5 | **94.8 ± 8.2** |
| **Average** | 38.4 | 54.4 | 65.1 | 77.0 | 45.8 | 62.1 | 54.8 | 89.4 | 90.1 | 75.2 | **94.5** |

of Maze2D and AntMaze, which require long-horizon reasoning. In these domains, our QDFFDT again demonstrates superior performance, leveraging sequence modeling for long-term dependencies while incorporating local signals to mitigate the impact of suboptimal sub-trajectories. Notably, our architecture achieves this without requiring explicit goal-conditioning, a prerequisite for methods like LSDT and QCS. The significant gains over purely return-conditioned models like LSDT and DT underscore the importance of integrating value learning, particularly in environments with sparse rewards. Finally, our QDFFDT demonstrates strong performance on the high-dimensional Kitchen and Adroit domains. This success in complex, manipulation-focused environments highlights the robustness and generalization capabilities of our hybrid approach.

## 6.2 COMPONENT ABLATION STUDY

To assess the effectiveness of our hybrid architecture, we conduct an ablation study on the individual feature extraction branches across D4RL and Atari domains (Mnih et al., 2013). For RCSL methods, we compare a pure sequence modeling baseline (DT), a purely Markovian return-conditioned

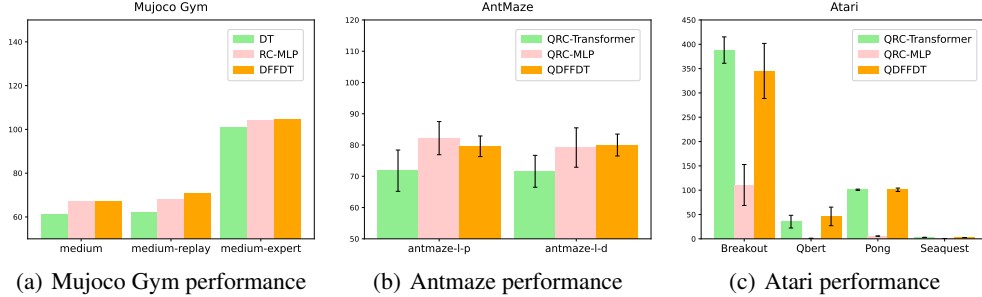

(a) Mujoco Gym performance  (b) Antmaze performance  (c) Atari performance

Figure 4: Ablation study on the contributions of the sequential and immediate feature branches. We report the performance of the sequential-only, immediate-only, and fused architectures. Average normalized scores are computed over 5 seeds for the Gym and AntMaze datasets, and 3 seeds for the Atari datasets.

MLP (RC-MLP), and our proposed DFFDT. For the value-augmented variants, we evaluate three models built upon a unified codebase to ensure fair comparisons. First, QRC-Transformer, our re-implementation of QT, serves as the pure sequence modeling baseline. Second, to establish a purely Markovian baseline, we introduce QRC-MLP, which replaces the Transformer backbone of QRC-Transformer with a simple MLP, thereby relying only on the current state. Finally, we present our QDFFDT, which integrates both approaches. As shown in Figure 4, on fully observable tasks such as Gym and AntMaze, pure sequence modeling tends to suffer from suboptimal segments within historical trajectories. This issue degrades both trajectory stitching and local feature extraction. In contrast, MLP-based methods that emphasize immediate information show stronger performance. On the Atari benchmark, high-dimensional visual observations need to be compressed into latent feature vectors for downstream RL. However, this compression process often results in significant information loss, making the tasks partially observable. In this scenario, sequence modeling becomes especially effective by leveraging historical context to compensate for the missing information. Although sequence modeling may be affected by suboptimal sub-trajectories in historical data, its strong capability to recover lost signals significantly outweighs this drawback. As a result, it demonstrates superior feature extraction compared to methods that rely solely on local observations. Our hybrid approach integrates both sequence modeling and local feature extraction, resulting in consistently strong performance across all benchmarks.

Table 3: The effects of $\hat{\alpha}$ in our QDFFDT. Average and standard deviation scores are reported over 5 seeds for Maze2D and AntMaze tasks. The best average values are marked in bold.

| | Maze2D | | AntMaze | | | | Average |
|---|---|---|---|---|---|---|---|
| | u | m | m-p | m-d | l-p | l-d | |
| QDFFDT($\hat{\alpha} = 0$) | $25.9 \pm 55.3$ | $144.5 \pm 17.3$ | $94.2 \pm 2.9$ | $95.4 \pm 0.8$ | $82.2 \pm 5.3$ | $79.2 \pm 6.3$ | 86.9 |
| QDFFDT($\hat{\alpha} = 0.25$) | $17.5 \pm 41.3$ | $117.1 \pm 22.8$ | $94.4 \pm 2.5$ | $94.4 \pm 1.0$ | $76.6 \pm 5.5$ | $82.2 \pm 4.3$ | 80.4 |
| QDFFDT($\hat{\alpha} = 0.5$) | $19.0 \pm 25.8$ | $132.5 \pm 22.0$ | $94.6 \pm 3.0$ | $95.2 \pm 1.0$ | $\mathbf{84.8} \pm 2.9$ | $\mathbf{84.0} \pm 3.4$ | 85.0 |
| QDFFDT($\hat{\alpha} = 0.75$) | $70.7 \pm 32.3$ | $145.8 \pm 9.9$ | $\mathbf{96.0} \pm 1.3$ | $94.2 \pm 1.2$ | $80.6 \pm 6.8$ | $82.0 \pm 2.3$ | 94.9 |
| QDFFDT($\hat{\alpha} = 1$) | $117.4 \pm 7.0$ | $156.6 \pm 2.5$ | $95.0 \pm 1.6$ | $94.8 \pm 1.5$ | $71.8 \pm 6.6$ | $71.6 \pm 5.1$ | 101.2 |
| QDFFDT | $\mathbf{123.3} \pm 19.7$ | $\mathbf{157.1} \pm 4.0$ | $94.8 \pm 2.0$ | $\mathbf{97.0} \pm 1.4$ | $79.6 \pm 3.3$ | $80.0 \pm 3.5$ | $\mathbf{105.3}$ |

## 6.3 Dynamic vs. Fixed Coefficient $\hat{\alpha}$

We have demonstrated the importance of integrating both sequential and Markovian features to support effective decision-making. In this subsection, we further examine the influence of dynamic versus static fusion coefficients. Within our dual-feature fusion framework, setting the fusion coefficient $\hat{\alpha}$ to 1 forces the policy to rely exclusively on sequence modeling, whereas setting $\hat{\alpha}$ to 0 makes it depend entirely on immediate information derived from the current state and return. We conduct experiments on our QDFFDT variants, evaluating static fusion settings by fixing $\hat{\alpha}$ to the set $\{0, 0.25, 0.5, 0.75, 1\}$, and report the results for Maze2D and AntMaze tasks in Table 3. The results

indicate that, with appropriate tuning, static coefficients can yield strong performance on specific datasets. However, achieving optimal performance requires careful manual tuning for each dataset, which reduces the general applicability of this approach. In contrast, our dynamically learned fusion coefficient consistently achieves robust performance across all datasets and obtains a higher average score than any fixed setting. These results highlight the effectiveness of the proposed adaptive fusion mechanism in combining global and local information without the need for extensive hyperparameter tuning.

## 7 CONCLUSION

In this work, we propose the Q-Augmented Dual-Feature Fusion Decision Transformer (QDFFDT), which combines sequential and immediate information through a learnable fusion mechanism. By dynamically adjusting the contributions of global and local features, our QDFFDT effectively alleviates the trajectory stitching limitations of conventional sequence modeling. Additionally, the integration of the Q-learning module further enhances value estimation and policy optimization. Extensive experiments on the D4RL benchmarks demonstrate that our QDFFDT consistently achieves superior performance across a range of tasks, including those with sparse rewards and high-dimensional state space, underscoring the effectiveness of value-guided feature fusion in offline RL.

## 8 REPRODUCIBILITY STATEMENT

We have made every effort to ensure the reproducibility of our work. Pseudocode for the proposed method is presented in Appendix A.1, and all hyperparameter settings are included in Appendix A.3. The datasets used in our experiments are publicly available through the D4RL benchmark (Fu et al., 2020).

ACKNOWLEDGMENTS

This work is supported in part by the National Natural Science Foundation of China under Grant 62471064; in part by the Guangxi Call-for-Bid Science and Technology Program JB2504240002; in part by the Fundamental Research Funds for the Beijing University of Posts and Telecommunications under Grant 2025AI4S02.

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

## A   IMPLEMENTATION DETAILS

### A.1   QDFFDT

Our QDFFDT follows the framework of DT (Chen et al., 2021), as shown in Algorithm 1.

---

**Algorithm 1** QDFFDT: Q-Augmented Dual-Feature Fusion Decision Transformer

---

Initialize V network $V_\psi$ and Alpha network $\alpha_\omega$.
*# Train the V network*
**for** each iteration **do**
    Sample transition mini-batch $\mathcal{B} = \{(\hat{R}_j, s_j)\} \sim \mathcal{D}$.
    Update $V_\psi$ by Equation 2.
**end for**.
*# Train the Alpha network*
**for** each iteration **do**
    Sample transition mini-batch $\mathcal{B} = \{(\hat{R}_j, s_j)\} \sim \mathcal{D}$.
    Update $\alpha_\omega$ by Equation 3.
**end for**.
Initialize policy network $\pi_\theta$ and critic networks $Q_{\phi_1}, Q_{\phi_2}$.
Initialize target networks $\pi_{\theta'}, Q_{\phi'_1}, Q_{\phi'_2}$.
*# Train the QDFFDT*
**for** each iteration **do**
    Sample sequence transition mini-batch $\mathcal{B} = \{(\hat{R}_j, s_j, r_j)_{j=t}^{t+K}\} \sim \mathcal{D}$.
    *# Q-value function learning*
    Sample target actions $\hat{a}$ via one of two methods:
        (n-step): Sample $\hat{a}_{t+K} \sim \pi_{\theta'}(\hat{a}_{t+K}|\hat{R}_{t+1:t+K}, s_{t+1:t+K})$
        (1-step): For each step $m \in [1, K]$, sample $\hat{a}_{t+1:t+i} \sim \pi_{\theta'}(\hat{a}_{t+1:t+i}|\hat{R}_{t+1:t+i}, s_{t+1:t+i})$
    Update $Q_{\phi_1}$ and $Q_{\phi_2}$ by Equation 7.
    *# Policy learning*
    **for** $i = 0$ to $K - 1$ **do**
        Sample $a_{t:t+i} \sim \pi_{\theta'}(a_{t:t+i}|\hat{R}_{t:t+i}, s_{t:t+i})$.
    **end for**.
    Update policy by minimizing Equation 8.
    *# Update target networks*
    $\theta' = \rho\theta' + (1 - \rho)\theta, \phi'_i = \rho\phi'_i + (1 - \rho)\phi_i$ for $i = \{1, 2\}$.
**end for**.
*# Inference with QDFFDT*
Given multiple target return-to-go choice $\hat{R}_0^{1:m}$ and initial state $s_0$.
**repeat**
    Sample multiple actions with different return-to-go $\tilde{a}_t^i = \pi_\theta(\tilde{a}_t^i|\hat{R}_{t-K+1:t}^i, s_{t-K+1:t})$ for $i = 1, \ldots, m$.
    Compute Q value with candidate state-action pair $(s_t, \tilde{a}_t^i)$ for $i = 1, \ldots, m$.
    Sample the action $a_t$ from action set $\{\tilde{a}_t^i\}_{i=1}^m$ with the max Q value by $\arg\max_{\tilde{a}_t^i} \min_{i=1,2} Q_{\phi'_i}(s_t, \tilde{a}_t^i)$.
    Execute the action $a_t$ and collect the reward $r_t$ and next state $s_{t+1}$.
    Update current return-to-go $\hat{R}_{t+1}^i = \hat{R}_t^i - r_t$ for $i = 1, \ldots, m$.
**until** $Done$ is $true$.

---

### A.2   NETWORK ARCHITECTURES

**V Network.** We build the V network as a 3-layer MLP with ReLU (Nair & Hinton, 2010) activation functions and 256 hidden units in each layer.

**Alpha Network.** We build the Alpha network as a 2-layer MLP with a ReLU activation and a Sigmoid output layer, using 256 hidden units.

**Q Networks.** We build two Q networks with the same architecture, each consisting of a 4-layer MLP with Mish activation functions and 256 hidden units in each layer.

**Conditional MLP Policy.** We build our RTG-conditioned MLP branch as a 3-layer MLP with GeLU (Hendrycks & Gimpel, 2016) activation functions and 256 hidden units in each layer.

**Conditional Transformer Policy.** We build our RTG-conditioned Transformer branch following the architecture of DT (Chen et al., 2021). To simplify integration and reduce implementation complexity, we utilize the codebase from `https://github.com/tinkoff-ai/CORL` instead of the official DT implementation.

## A.3 HYPERPARAMETERS

Table 4: Hyperparameters of QDFFDT in our experiments.

| Parameter | Value |
|---|---|
| Number of layers | 3 |
| Number of attention heads | 1 |
| Embedding dimension | 512 Kitchen and Adroit |
| | 256 other tasks |
| Nonlinearity function | GeLU |
| Grad norm clip | 5.0 |
| Batch size | 256 |
| Dropout | 0.1 |
| Optimizer | Adam (Kingma, 2014) |
| Learning rate | 3e-4 |
| Temperature coefficient | 1e5 |
| $\alpha_{\min}$ | 0.3 |

For the Gym domain, we use a context length of 20, while a context length of 5 is used for the other domains. The discount factor is set to $\gamma = 0.99$ by default, and increased to $\gamma = 0.995$ for AntMaze tasks, following the design choice in SPOT (Wu et al., 2022). The expectile level is typically set to $\sigma = 0.7$, except for the *antmaze-large* datasets, where we set it to 0.9. For Q-value updates, we employ the 1-step Bellman equation on *halfcheetah* datasets and the n-step Bellman equation on all other tasks. Additional detailed hyperparameter settings for actor training are given in Table 4. Finally, $\lambda$ is adjusted according to the characteristics of different datasets. For example, *medium-expert* datasets usually require more policy regularization due to their high quality, while *medium* and *medium-replay* tasks require more Q-learning to improve the policy. Based on these considerations, the choice of $\lambda$ for each dataset is shown in Table 5.

## B BASELINE DETAILS

We evaluate the performance of QDFFDT with many baselines. These methods consist of traditional value-based methods: CQL (Kumar et al., 2020), IQL (Kostrikov et al., 2021b), BEAR Kumar et al. (2019), AWR (Peng et al., 2019), TD3+BC (Fujimoto & Gu, 2021), EQL (Xu et al., 2023), LAPO (Chen et al., 2022), and CQL+AW (Hong et al., 2023); RCSL and its derivative methods: DT (Chen et al., 2021), DC (Kim et al., 2023), LSDT (Wang et al., 2025), RvS (Emmons et al., 2021), GDT (Hu et al., 2023), QDT (Yamagata et al., 2023), EDT (Wu et al., 2024), Reinformer (Zhuang et al., 2024), POR (Xu et al., 2022), CGDT (Wang et al., 2024), ACT (Gao et al., 2024), QT (Hu et al., 2024), and QCS (Kim et al., 2024); as well as generative-model-based methods: BCQ (Fujimoto et al., 2019), A2PO (Qing et al., 2024), Diffuser (Janner et al., 2022), DD (Ajay et al., 2022), and D-QL (Wang et al., 2022). The performance results for these baselines are taken from their original publications, with the exception of QT. In the original QT paper, scores on the AntMaze and Adroit domains are averaged over 3 random seeds with 10 evaluations per dataset, which is inconsistent with the evaluation protocol used by other baselines. To ensure fair comparison, we re-evaluate QT using the official codebase, computing average performance over 100 evaluations for every seed.

Table 5: Hyperparameter settings of all selected tasks.

| Tasks | $\lambda$ |
|---|---|
| halfcheetah-medium-expert-v2 | 0.1 |
| hopper-medium-expert-v2 | 1.0 |
| walker2d-medium-expert-v2 | 0.5 |
| halfcheetah-medium-v2 | 0.01 |
| hopper-medium-v2 | 0.2 |
| walker2d-medium-v2 | 0.2 |
| halfcheetah-medium-replay-v2 | 0.01 |
| hopper-medium-replay-v2 | 0.2 |
| walker2d-medium-replay-v2 | 0.2 |
| maze2d-umaze-v1 | 0.3 |
| maze2d-medium-v1 | 0.2 |
| maze2d-large-v1 | 0.1 |
| antmaze-umaze-v0 | 0.5 |
| antmaze-umaze-diverse-v0 | 0.5 |
| antmaze-medium-play-v0 | 0.3 |
| antmaze-medium-diverse-v0 | 0.3 |
| antmaze-large-play-v0 | 0.2 |
| antmaze-large-diverse-v0 | 0.2 |
| kitchen-partial-v0 | 50.0 |
| kitchen-mixed-v0 | 20.0 |
| pen-human-v1 | 8.0 |
| pen-cloned-v1 | 9.0 |

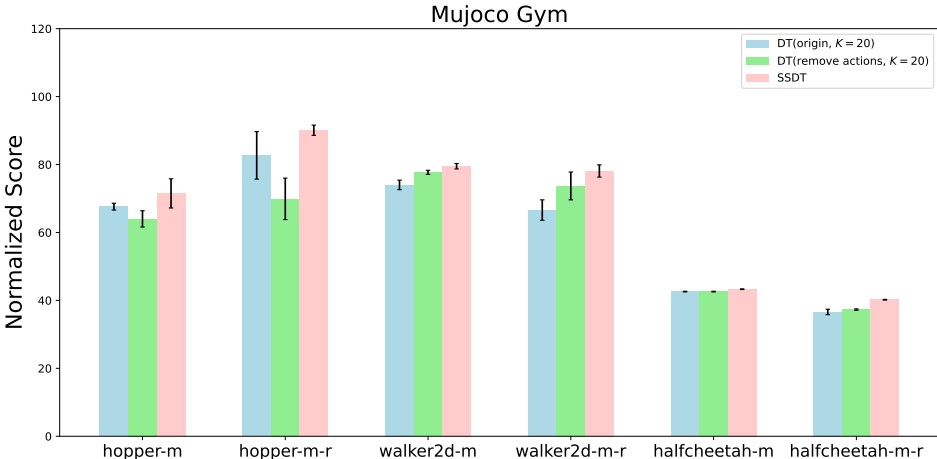

Figure 5: Evaluation results for DT, DT (remove actions), and SSDT on the MuJoCo Gym suboptimal datasets. We record the normalized scores averaged across 3 seeds, 10 evaluations each.

## C  DISCUSSIONS OF AUXILIARY EXPERIMENTS

### C.1  COMPARATIVE ANALYSIS OF SSDT AND DT

To further investigate the role of sequence length, particularly the weaknesses of long-term context on suboptimal datasets as discussed in Section 2, we conduct a comparative analysis between SSDT and DT on the MuJoCo Gym benchmarks. Since single-step models make decisions based solely

on the current state and target return without action inputs, we establish a fair baseline by using a DT variant with action tokens removed. To ensure equivalent data exposure across models with varying sequence lengths, we adjust the number of training iterations: 25,000 for a sequence length of 20 and 500,000 for a sequence length of 1 (e.g., SSDT). All other hyperparameters, including the batch size of 256, align with the official DT implementation. As shown in Figure 5, SSDT consistently outperforms the standard DT across all tested tasks. This result is particularly insightful because it challenges the conventional assumption that longer temporal context is always beneficial. The strong performance of SSDT stems from the fully observable, Markovian nature of the Gym environments. In such a setting, the current state theoretically contains all information required for optimal decision-making. Consequently, for DT, conditioning on a long context can introduce a restrictive inductive bias, potentially forcing the model to replicate suboptimal historical patterns rather than selecting the optimal action from the current state. SSDT, with its context length of 1, effectively avoids this bias. It is crucial to note that this finding does not diminish the importance of sequence modeling in general. In Partially Observable MDPs (POMDPs), such as Atari from pixels, history is vital for inferring the latent state from ambiguous single observations (Bhargava et al., 2023).

## C.2 EXPERIMENTAL DETAILS OF DFFDT

Our DFFDT framework (Section 6.2) is implemented on top of the original DT, and therefore inherits the majority of its hyperparameter settings. The major modifications concern the introduction of additional training hyperparameters for the dynamic weighting coefficient. A complete list of these settings is provided in Table 6.

Table 6: Hyperparameters of DFFDT in our experiments.

| Parameter | Value |
|---|---|
| Number of layers | 3 |
| Number of attention heads | 1 |
| Embedding dimension | 128 |
| Nonlinearity function | ReLU |
| Grad norm clip | 0.25 |
| Batch size | 64 |
| Context length | 20 |
| Dropout | 0.1 |
| Learning rate | 1e-4 |
| Temperature coefficient | 1e5 |
| $\alpha_{\min}$ | 0.3 |

## C.3 EXPERIMENTAL DETAILS ON ATARI

To illustrate the impact of sequence-level representations in pixel-based environments, we conduct a comparative study on the Atari domain between the sequence modeling approach QRC-Transformer, the single-step modeling variant QRC-MLP, and our hybrid method QDFFDT in Section 6.2. Because the performance gains afforded by sequence modeling in image-based scenarios substantially outweigh drawbacks introduced by suboptimal historical sub-trajectories, we maintain a high proportion of sequence modeling by manually tuning the feature-weighting coefficient $\hat{\alpha}$. Additionally, to mitigate the burden of computational resources, we reduce both the number of attention heads and the number of layers in the sequence-modeling module. Final scores are normalized following (Hafner et al., 2020). The complete hyperparameter configurations are detailed in Table 7.

Table 7: Hyperparameters of QDFFDT for Atari experiments.

| Hyperparameter | Value |
|---|---|
| Number of layers | 3 |
| Number of attention heads | 1 |
| Batch size | 128 Pong |
| | 64 other tasks |
| Context length | 50 Pong |
| | 30 other tasks |
| Embedding dimension | 256 |
| Nonlinearity | ReLU, encoder |
| | GeLU, otherwise |
| Encoder channels | 32, 64, 64 |
| Encoder filter sizes | $8 \times 8, 4 \times 4, 3 \times 3$ |
| Encoder strides | 4, 2, 1 |
| Max epochs | 10 |
| Dropout | 0.1 |
| Learning rate | $6 \times 10^{-4}$ |
| Adam betas | (0.9,0.95) |
| Grad norm clip | 1.0 |
| Weight decay | 0.1 |
| Learning rate decay | Linear warmup and cosine decay |
| Warmup tokens | $512 * 20$ |
| Final tokens | $2 * 500000 * K$ |
| $\hat{\alpha}$ | 0.9 Pong |
| | 1.0 other tasks |

## D    SUPPLEMENTARY STUDIES

### D.1    COMPARISON BETWEEN ADDITION-BASED FUSION AND CONCATENATION FUSION

To evaluate our adaptive addition-based fusion module (Add) against the concatenation-based fusion module (Cat), we combine contextual and Markovian features by concatenation followed by a linear projection, yielding

$$h = \text{Concat}\big((1 - \hat{\alpha}(s_t)) \cdot h_t^{loc}; \hat{\alpha}(s_t) \cdot h_t^{glo}\big), \tag{9}$$

where $h_t^{loc}$ and $h_t^{glo}$ denote the single-step and contextual representations (see Section 5.1.2). Under identical hyperparameters on the *walker2d-medium-replay-v2* and *hopper-medium-replay-v2* datasets, Add and Cat achieve closely matched performance (Table 8). Since Add requires slightly fewer parameters, we adopt it as our preferred fusion strategy.

Table 8: Performance comparison between addition-based fusion and concatenation-based fusion. Average and standard deviation scores are reported over 5 seeds. The best average values are marked in bold.

| Dataset | Add | Cat |
|---|---|---|
| walker2d-medium-replay-v2 | **101.0** $\pm$ 1.3 | 100.3 $\pm$ 1.8 |
| hopper-medium-replay-v2 | 102.7 $\pm$ 0.5 | **103.9** $\pm$ 0.3 |

### D.2    ABLATION STUDY ON EXPECTILE LEVEL $\sigma$

We incorporate expectile regression to estimate an upper bound of RTGs and use the discrepancy between this estimate and the observed RTGs during training to dynamically adjust the weighting between sequential and single-step modeling. To understand the effect of the expectile level $\sigma$, we conduct an ablation study. As illustrated in Figure 6, varying $\sigma$ influences the relative contribution

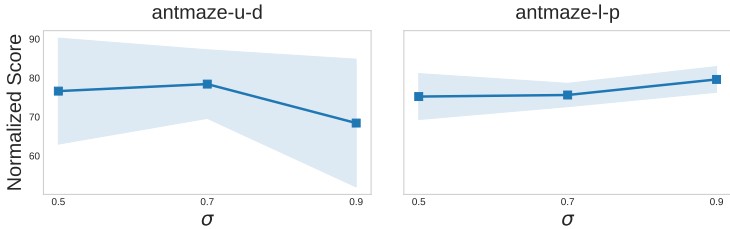

Figure 6: Ablation on the $\sigma$. Each line represents the variation in the average scores reported over 5 seeds, and the shaded area represents the corresponding standard deviation.

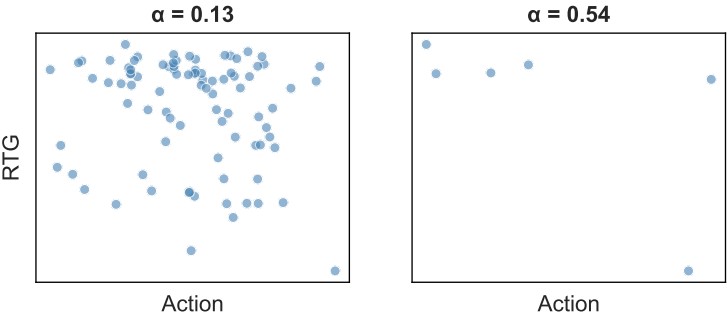

Figure 7: Visualization of the correlation between the $\alpha$ and the RTG distribution. Each subfigure corresponds to a different fixed state. The x-axis represents the 1D projection of action vectors via PCA..

of sequential features. Higher values of $\sigma$ reduce the weight assigned to sequential modeling, which can help mitigate the negative impact of suboptimal sub-trajectories in historical data. However, in partially observable environments such as AntMaze, sequential information provides valuable context for decision-making. Therefore, selecting an appropriate value of $\sigma$ is critical for achieving a favorable trade-off between leveraging historical context and mitigating the influence of suboptimal behavioral trajectories.

### D.3 RELATIONSHIP BETWEEN $\alpha$ AND RTG DISTRIBUTION DURING TRAINING

Single-mode approaches face fundamental limitations. Pure sequence models can reproduce suboptimal sub-trajectories by overemphasizing historical information, whereas purely local models tend to fit the average of expert actions. To address these issues, we introduce the dynamically adjusted weighting factor $\hat{\alpha}$ to control the fusion of global and local features. The final fusion coefficient $\hat{\alpha}$ is obtained by scaling the output of the Alpha Network (Section 5.1.1), and the raw output $\alpha$ is positively correlated with $\hat{\alpha}$. Analyzing $\alpha$ thus provides an effective lens to understand feature fusion behavior. Figure 7 visualizes the distribution of $\alpha$ on the *walker2d-medium-v2* dataset. For RTG distributions with larger or more frequent fluctuations, $\alpha$ tends to be lower, indicating reduced reliance on sequence modeling. Our fusion mechanism mitigates the limitations of single-mode models by preserving sequence stability on high-reward trajectories while adaptively amplifying local signals in suboptimal regions, thereby enabling more effective trajectory stitching.

### D.4 ANALYSIS OF THE Q-VALUE UPDATE STRATEGY

As described in Appendix A.3, we apply the 1-step Bellman equation to the *halfcheetah* datasets. This choice is motivated by our observation that model performance degrades when using the n-step Bellman backup in combination with longer context lengths. Since we incorporate Markovian local features into the fused representation, which are independent of sequence length, we hypothesize that this degradation stems from the n-step Bellman backup's dependence on trajectories. To verify

Table 9: Performance with varying context length $K$ on the *halfcheetah-medium-v2*. Average and standard deviation scores are reported over 5 seeds.

|  | $K$=5 | $K$=10 | $K$=20 |
|---|---|---|---|
| n-step Bellman | $65.2 \pm 1.1$ | $59.6 \pm 0.6$ | $55.6 \pm 0.9$ |
| 1-step Bellman | $66.9 \pm 1.3$ | $64.7 \pm 1.5$ | $65.7 \pm 1.3$ |

Table 10: Performance with varying context length $K$ on the *halfcheetah-medium-replay-v2*. Average and standard deviation scores are reported over 5 seeds.

|  | $K$=5 | $K$=10 | $K$=20 |
|---|---|---|---|
| n-step Bellman | $55.8 \pm 1.7$ | $48.3 \pm 0.7$ | $38.4 \pm 1.1$ |
| 1-step Bellman | $64.2 \pm 0.8$ | $62.8 \pm 1.8$ | $62.0 \pm 1.1$ |

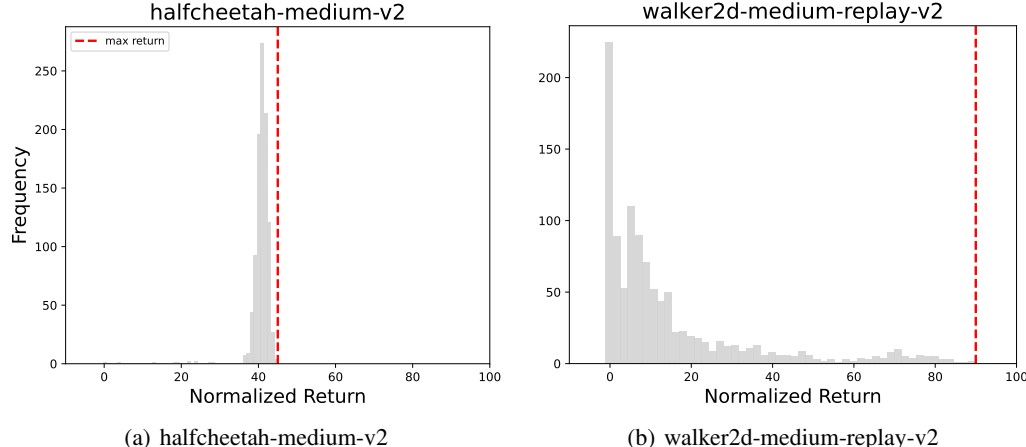

(a) halfcheetah-medium-v2          (b) walker2d-medium-replay-v2

Figure 8: The distribution of returns of trajectories for *halfcheetah-medium-v2* and *walker2d-medium-replay-v2* datasets. The dashed lines represent the maximum trajectory returns.

this, we conduct an ablation study on *halfcheetah-medium-v2* and *halfcheetah-medium-replay-v2*. Table 9 and Table 10 demonstrate that increasing sequence length impairs trajectory stitching when using the n-step Bellman equation, whereas performance remains stable under the 1-step Bellman equation, thereby exposing a limitation of n-step methods in certain settings. However, QT (Hu et al., 2024) demonstrates that n-step updates outperform 1-step updates on *walker2d-medium-replay-v2*. To illustrate this discrepancy, we plot the normalized return distributions for *halfcheetah-medium-v2* and *walker2d-medium-replay-v2* in Figure 8. On *halfcheetah-medium-v2*, the return distribution is concentrated around 40, indicating a large number of medium-return trajectories. These dense, moderate-reward sequences provide sufficient local learning signals that reduce the impact of bias in 1-step Bellman updates. However, due to the absence of long, contiguous high-return trajectories, the n-step Bellman targets are often underestimated, as the cumulative rewards over longer horizons remain limited. This leads to degraded performance when n-step updates are applied. In contrast, *walker2d-medium-replay-v2* presents a long-tailed return distribution with a predominance of low-return trajectories and sparse but critical high-return segments. In this setting, the limited coverage of high-return regions prevents 1-step updates from accurately propagating value signals, resulting in pronounced estimation bias. N-step backups, by aggregating rewards across extended time spans, help bridge these gaps and anchor credit assignment in high-return regions, thereby mitigating the shortcomings of 1-step updates and enabling more effective policy learning. These observations underscore the distinct advantages of each strategy in different scenarios: 1-step updates are well-suited for environments with locally concentrated return distributions, whereas n-step backups are

more effective in long-tailed settings where high-return trajectories are sparse and require longer temporal credit assignment.

## E    TRAINING TIME

We compare the training efficiency of our proposed methods, DFFDT and QDFFDT, against the baseline methods, DT and QT. All experiments are conducted on a single NVIDIA RTX 2080 Ti GPU. In terms of training time, DFFDT requires 1.19 hours, while DT takes 0.79 hours. QDFFDT takes 3.88 hours compared to 3.59 hours for QT. Regarding memory usage, the peak GPU memory consumption is 1.226 GB for DFFDT, 1.216 GB for DT, 2.836 GB for QDFFDT, and 2.742 GB for QT. To ensure fairness, QDFFDT and QT use identical hyperparameters wherever applicable, and the same configuration is adopted for DFFDT and DT. These results show that the additional computational cost introduced by our feature fusion design is limited, while the performance gains are substantial.

## F    LIMITATIONS

Our QDFFDT adaptively fuses global and local features based on the estimated quality of future trajectories, with a Q-learning component to guide policy improvement. Experimental results across a range of tasks demonstrate its effectiveness in trajectory planning. However, the overall architecture introduces additional structural complexity compared to standard sequence modeling approaches. Reducing this complexity while maintaining or improving performance remains an important direction for future work.

## G    FUTURE WORK

Several promising directions remain for future exploration. First, incorporating multi-scale temporal modeling (Qiu et al., 2025) could enhance credit assignment across varying time horizons. Second, investigating compositional task structures (Cao et al., 2025a) may improve generalization in multi-task offline settings. Finally, exploring tree-based policy optimization (Cao et al., 2025b) for hierarchical decision making remains a promising direction.

## H    THE USE OF LARGE LANGUAGE MODELS (LLMS)

In accordance with the ICLR policy on the use of large language models (LLMs), we disclose that LLMs are employed during the preparation of this paper. Their use is limited to assisting with language editing, improving readability, and polishing the presentation of the manuscript. LLMs are not involved in formulating the research questions, designing the methodology, conducting experiments, analyzing results, or drawing scientific conclusions. We take full responsibility for all technical content, claims, and conclusions presented in this work.

