# OpenReview forum: "Offline Reinforcement Learning with Adaptive Feature Fusion"
_ICLR.cc/2026/Conference — ICLR 2026 Poster_

### Official Review · Reviewer_n7e3 · 2025-10-25

**Soundness:** 2
**Presentation:** 2
**Contribution:** 2
**Rating:** 4
**Confidence:** 4

**Summary:**

This paper introduces QDFFDT and DFFDT, novel offline RL models that mitigates the limitations of purely sequence-based RCSL. DFFDT fuses global sequential and local immediate features through a learnable fusion weight, enabling effective balance between long-term context and Markovian decision signals. QDFFDT further integrates Q-learning to enhance value estimation and policy improvement. Experiments on the D4RL benchmark demonstrate that QDFFDT achieves SOTA across diverse domains.

**Strengths:**

The paper presents a clear illustration of the limitations of prior return-conditioned sequence models and introduces an adaptive transformer architecture that empirically achieves strong performance across diverse offline RL tasks.

**Weaknesses:**

The paper’s contribution primarily lies in modifying the architecture of the Decision Transformer for sequential decision modeling. The authors aim to learn a mapping $f(a_t \mid \hat{R}_{\leq t}, s_{\leq t})$ and introduce a dual-feature fusion mechanism to encourage attention to immediate Markovian information. However, this design raises a theoretical concern:

The input information remains unchanged, and a sufficiently expressive Transformer should, in principle, be capable of learning any desired feature weighting — including the adaptive combination $((1-\hat{\alpha}(s_t))h^{loc}_t + \hat{\alpha}(s_t)h^{glo}_t)$ — directly through its self-attention and learned parameters. The additional fusion module, therefore, seems architecturally redundant unless the authors can demonstrate a theoretical advantage.

**Questions:**

1. I suggest that the authors improve the Abstract and Introduction to define the problem and present their motivation more clearly. For example, in the Abstract, they state that existing approaches “lead to an overreliance on suboptimal past experience,” which is not immediately clear to the reader. In the Introduction, the authors should describe the problem more explicitly in the second paragraph and clarify the method and its rationale in the third paragraph. They may also consider moving Section 4 into the Introduction.

2. It would be helpful to include the Q-component in Figure 3. With this component, the architecture corresponds to QDFFDT; without it, it represents DFFDT.

---

> ### Author Response · Authors · 2025-11-22
>
> Thank you for your thoughtful review of our research. We appreciate your constructive feedback and hope our suggested changes and this individual response will address your concerns in detail:
>
> **Weakness:**
>
> We agree in principle that a sufficiently powerful Transformer possesses the theoretical expressivity to learn any complex feature weighting. However, we respectfully argue that there is a significant and crucial gap between this theoretical potential and the practical learning dynamics within the challenging context of offline RL.
>
> The core issue is that a standard Transformer architecture, being a general-purpose sequence model, lacks a specific **inductive bias** for Markov Decision Processes (MDPs). When trained on a fixed dataset of trajectories, it defaults to modeling the joint distribution of the entire sequence. This makes it highly susceptible to overfitting to long-term correlations within suboptimal trajectories, as it has no inherent mechanism to distinguish between locally good decisions and the suboptimal global context in which they appear.
>
> Our proposed dual-feature fusion module is not intended to increase the model's raw expressive power, but rather to introduce a critical **structural inductive bias** that aligns the learning process with the properties of sequential decision-making. By explicitly separating local (Markovian) and global (historical) information streams, our module reframes the learning problem. Instead of asking the model to discover the principle of Markovian locality from scratch, we simplify its task to learning an adaptive weighting between these two streams. This explicit structural guidance is vital for preventing the model from over-relying on misleading historical context.
>
> Therefore, we respectfully argue the module is not "architecturally redundant" but is a necessary mechanism to compensate for the standard Transformer's lack of domain-specific priors for decision-making. Our strong empirical results, which show a significant performance gain with this module, serve as validation that this explicit bias is necessary in practice and is not something the standard Transformer architecture learns on its own from the available offline data.
>
> **Q1:**
>
> Thank you for these excellent and detailed suggestions for improving the paper's clarity and structure. We agree that presenting a stronger, more explicit motivation at the beginning would significantly benefit the reader. Following your advice, we have performed a comprehensive revision of the Abstract and Introduction. To make these changes easy for you to identify, we have highlighted all new or significantly revised text with \textcolor{blue} in the updated PDF of our manuscript.
>
> Specifically, we have:
>
> 1. **Revised the Abstract** to more clearly define the core problem of "overreliance on suboptimal past experiences" in the context of failed trajectory stitching.
> 2. **Restructured the Introduction** to immediately establish this problem, present our core insight, and then introduce our method as the solution.
>
> You also made a very good point about integrating our motivational case study (originally Section 4) earlier. We found that placing the entire case study, including its figure, table, and detailed analysis, directly inside the introduction made it overly long and disrupted its narrative flow. Therefore, we have adopted a structure that we believe achieves your goal effectively: we have moved the case study to become the new Section 2, titled "Motivation: When Pure Sequence Modeling Hinders Trajectory Stitching". This places the motivation immediately after the introduction, giving this critical argument the space and prominence it deserves while maintaining a clean, readable structure for the paper. We are confident that these revisions have substantially improved the paper's clarity and impact. We kindly invite you to review the updated manuscript.
>
> **Q2:**
>
> Our intention with Figure 3 was to provide a focused illustration of the core architectural novelty of our work: the Dual-Feature Fusion Decision Transformer (DFFDT) policy network. As the caption for Figure 3 already states, 'An overview of our Dual-Feature Fusion Decision Transformer architecture', the diagram is specifically dedicated to this new component.
> The Q-learning component, which completes our full QDFFDT model, follows the well-established TD3+BC paradigm. It is a common and standard practice in the offline RL literature to present the novel policy architecture visually, while detailing the integration of standard value-based components, like Q-functions, through the formal equations of the learning objective. We follow this convention by providing the precise mathematical formulation of the Q-learning integration in Section 5.2. We believe this approach effectively balances clarity and focus, allowing us to highlight our primary contribution without cluttering the diagram with standard elements.

---

### Official Review · Reviewer_EsME · 2025-10-29

**Soundness:** 4
**Presentation:** 4
**Contribution:** 4
**Rating:** 8
**Confidence:** 4

**Summary:**

This paper proposes QDFFDT (Q-Augmented Dual-Feature Fusion Decision Transformer), an offline reinforcement learning framework that adaptively combines global sequence modeling and local single-step (Markovian) modeling through a learnable fusion mechanism. The method extends return-conditioned supervised learning (RCSL) by learning a state-dependent fusion coefficient based on the expectile regression of return-to-go values, balancing the influence of contextual and immediate features. Additionally, a Q-learning module is integrated to improve policy optimization and mitigate the overreliance on suboptimal trajectory patterns common in sequence-based methods like Decision Transformer (DT) and QT. Experiments on the D4RL benchmark across Gym-MuJoCo, Maze2D, AntMaze, Kitchen, and Adroit show that QDFFDT achieves consistent state-of-the-art performance, improving generalization without extensive hyperparameter tuning.

**Strengths:**

- Well-motivated insight: The paper identifies a real weakness in RCSL methods—overfitting to suboptimal sequence fragments that hinder trajectory stitching—and directly targets it with an adaptive fusion approach.

- Elegant hybrid design: The fusion of sequential and immediate features through a learned state-conditioned weight is conceptually neat and grounded in the Markov property. It’s a balanced architectural innovation rather than an ad hoc ensemble.

- Integration of value learning: The Q-augmentation complements RCSL’s generative bias, yielding both theoretical plausibility and empirical strength.

- Comprehensive experiments: The authors evaluate on five major domains (Gym, Maze2D, AntMaze, Kitchen, Adroit) against more than 15 baselines, consistently outperforming DT, QT, and RCSL variants. Ablations (static vs. adaptive α̂, sequence-only vs. single-step) are insightful and well-documented.

- Clarity and reproducibility: The paper is clearly written, with structured explanations, pseudo-code, and full hyperparameter tables. The method is easy to reimplement.

**Weaknesses:**

- Limited theoretical contribution: Despite clear empirical validation, the method is largely architectural. The theoretical motivation for how expectile-based weighting improves trajectory stitching remains heuristic.

- Overcomplexity relative to gain: The approach introduces five networks and multiple losses (expectile, α-loss, Q-loss, MSE) for modest algorithmic advancement; simplicity compared to QT or ACT is reduced.

- Incremental relationship to prior work: The approach closely resembles QT (Hu et al., 2024) with an extra fusion branch and expectile weighting. The novelty over these strong baselines feels evolutionary rather than revolutionary.

**Questions:**

1. How sensitive is QDFFDT to the expectile level σ and the temperature parameter T in α-learning?

2. Does α̂(s) exhibit meaningful variation across states or converge to near-constant values? Can this be visualized?

3. What is the runtime overhead compared to QT or DT in wall-clock training?

4. Would replacing the Transformer with a lightweight recurrent or convolutional encoder yield similar benefits?

---

> ### Author Response · Authors · 2025-11-22
>
> Thank you for your thoughtful review of our research. We appreciate your constructive feedback and hope our suggested changes and this individual response will address your concerns in detail:
>
> **Weaknesses (Contribution, Complexity, and Novelty):**
>
> We agree with your overall assessment of our work's contribution. Our primary goal was to provide a practical, architectural solution to a specific limitation in existing return-conditioned supervised learning (RCSL) methods, validated by strong empirical evidence.
>
> - **Theoretical Contribution:** We acknowledge that our work is primarily empirical. Our aim was to design a robust, well-motivated architecture, while ensuring that its theoretical underpinnings, such as convergence properties, are inherited from the established and well-studied frameworks of TD3+BC and QT.
> - **Novelty and Complexity:** Our work is indeed an evolutionary step, building directly on strong foundations like QT. Our goal was not to reinvent the paradigm, but to identify and solve a specific and critical limitation: the static and inflexible trade-off between global and local information in prior models. This targeted solution does introduce additional components, but we took care to ensure the practical overhead is minimal. As detailed in Appendix E and our response to Q3 below, the increase in training time and memory is marginal. We believe this is a worthwhile trade-off for the significant performance gains and improved adaptability our method provides.
>
> **Q1:**
>
> We have conducted new ablation studies on both $\sigma$ and $T$ to analyze their sensitivity. These two hyperparameters jointly control the learning of the adaptive fusion weight $\hat\alpha(s)$. Our results show that the model's performance is stable across a reasonable range for both hyperparameters.
>
> - **Sensitivity to Expectile Level $\sigma$:**
>
> | $\sigma$ | maze2d-umaze |
> | --- | --- |
> | 0.5 | 123.3 ± 10.4 |
> | 0.7  | 123.3 ± 19.7 |
> | 0.9 | 125.1 ± 12.1 |
> - **Sensitivity to Temperature $T$:**
>
> | $T$ | maze2d-umaze |
> | --- | --- |
> | 10,000 | 117.8 ± 27.1 |
> | 100,000 | 123.3 ± 19.7 |
> | 1,000,000 | 108.0 ± 20.1 |
>
> These results demonstrate that our method is robust to variations in these hyperparameters. We will add these ablation studies to the Appendix.
>
> **Q2:**
>
> As visualized in Figure 7 of our paper, $\hat\alpha(s)$ exhibits meaningful, state-dependent variation and does not converge to a constant.
> The learned $\hat\alpha(s)$ effectively acts as a dynamic confidence measure. In states where the distribution of RTG values across different trajectories has high variance (indicating that the state is part of both high- and low-quality outcomes in the dataset), $\hat\alpha(s)$ is driven lower. This correctly increases the model's reliance on the more robust, local Markovian features. Conversely, in states that are consistently part of high-return trajectories, $\hat\alpha(s)$ trends higher, allowing the model to trust the global sequence context.
>
> **Q3:**
>
> As detailed in Appendix E, the runtime and memory overhead of QDFFDT is minimal compared to baselines like QT and DT. While our model does have more components, they are lightweight, and the end-to-end training time is only marginally higher. We believe this small increase in computational cost is well justified by the performance improvements.
>
> **Q4:**
>
> The central contribution of our paper is the adaptive dual-feature fusion mechanism, not the specific choice of the Transformer as the backbone.
> We agree that this fusion principle could likely be successfully applied to other powerful sequential architectures. Our mechanism's goal of adaptively balancing historical context with immediate Markovian information is a principle relevant for any sequence model used in RL. As validated by other successful works like "Rethinking Transformers in Solving POMDPs"[1] (using RNNs) and "Decision convformer"[2] (using CNNs), the specific choice of the sequence encoder is often flexible.
> Therefore, our contribution is orthogonal to, and not in conflict with, research that focuses on different sequential backbones like RNNs or CNNs. The fusion mechanism we propose could be integrated with these architectures as well.
> While this is a promising direction for future work, we did not conduct these specific experiments in the current study. This was due to the lack of established, state-of-the-art baselines that combine Q-learning with RNN or CNN backbones in a manner directly comparable to QT. Properly developing and validating such architectures would be a substantial undertaking, beyond the scope and time constraints of the rebuttal period. Thank you for this suggestion.
>
> [1] Lu, Chenhao, et al. "Rethinking transformers in solving POMDPs." arXiv preprint arXiv:2405.17358 (2024).
>
> [2] Kim, Jeonghye, et al. "Decision convformer: Local filtering in metaformer is sufficient for decision making." arXiv preprint arXiv:2310.03022 (2023).

---

> > ### Comment · Reviewer_EsME · 2025-11-26
> >
> > Thank you for the thorough rebuttal. The authors have adequately addressed the key issues I previously noted, and the additional analysis and clarification strengthened the paper. I am satisfied with the responses, and my recommendation remains accept.

---

> > > ### Author Response · Authors · 2025-11-26
> > >
> > > We are pleased to know our response has clarified your concerns. Thank you once again for leaving a thoughtful comment and for dedicating your time and effort to reviewing our work.

---

### Official Review · Reviewer_qJYv · 2025-10-31

**Soundness:** 2
**Presentation:** 3
**Contribution:** 3
**Rating:** 4
**Confidence:** 4

**Summary:**

This paper combines long-term sequential modeling and local immediate modeling via an adaptive feature fusion mechanism for efficient offline return-conditioned supervised learning (RCSL). Previous pure sequence modeling methods tend to suffer from replicating suboptimal trajectories in the training dataset, and additional Q-value guidance cannot eliminate this drawback. By estimating an approximated max-return for each state, the proposed method DFFDT can adaptively rely more on the local one-step action prediction when suboptimal RTG signals are observed (after comparing them with the estimated V-values). With the aid of the Q value, the variant QDFFDT achieves superior performance compared to strong baselines on D4RL benchmarks and is claimed to be flexible in balancing the use of global and local features.

**Strengths:**

- The paper is well-written and easy to follow. Figure 3 efficiently captures the main idea of this paper. Notations are clean and consistent.
- The core component, feature fusion, is well-motivated through a necessary didactic example in Section 4 and demonstrates that the Q-value guidance alone is insufficient for counteracting the “momentum”  of replicating the suboptimal trajectories.
- QDFFDT performs well on D4RL benchmarks with an acceptable increase in training complexity (as shown in Appendix E).
- Appendix includes enough experimental details for reproduction purposes.

**Weaknesses:**

- **In terms of the didactic example,** why does QT perform worse when $\eta=10.0$ than when $\eta=1.0$? Increasing $\eta$ gradually eclipses the effect of the DT loss and enables QT to choose actions with the highest Q values (which is true for $\eta=0.01-1.0$). Intuitively, setting $\eta$ to a very large number should completely erase the effect of the BC term, and thus QT should be able to achieve a higher success rate.
- In my opinion, the core contribution of this paper is the content in Sections 4 and equations (3) and (5), which trains an Alpha network to adaptively trade off between global sequential features and local immediate features. **However, I have the following concerns about the effectiveness of feature fusion, despite QDFFDT’s strong performance shown in Table 2:**
    1. In terms of the training objective of the Alpha network, a naive solution to minimize it could predict $\alpha_\omega (s) = 0 $ for most of the state. And when $V_\psi (s) \leq \hat{R}$, there is no gradient signal to control the training behavior. I suspect that this adaptive mechanism would eventually bias towards learning from local features more than from global features. This suspicion is amplified, especially when the authors have to manually set $\hat{\alpha}=1$ for 3 of the 4 evaluated Atari games to ensure the model fully relies on the sequence modelling (since it benefits largely for these partially observed environments). Moreover, in Appendix D.3, it seems to be true that the learned $\alpha$ is small in general. Could the authors comment on this? What is the average learned $\alpha_\omega(s)$ for these datasets?
    2. In terms of the ablation study, Figure 4 gives me a feeling that (1) for environments with the Markovian property, local modeling alone is enough; and (2) for partially observed environments, global modeling alone is enough. The performance gain of feature fusion is not as prominent as in Table 2, if we take the variance into consideration. Given the question (a) above, it makes sense for (Q)DFFDT working relatively well for Gym and Antmaze, since it might predict low $\alpha$ most of the time. And what is the performance for Atari, if $\hat{\alpha}$ is not manually set?
    3. In summary, I feel that it is a bit far-fetched to conclude that “These results highlight the effectiveness of the proposed adaptive fusion mechanism in combining global and local information without the need for extensive hyperparameter tuning.” (Lines 473-475). The authors should provide more evidence to defend this core argument.
- **Some experimental details need more clarification:**
    1. In Table 2, why does QT perform poorly for antmaze-l, while the reimplementation of QT, QRC-Transformer, performs much better in Figure 4(b)?
    2. I am aware that most of the offline methods fail on Adroit-door, Adroit-hammer, and Adroit-relocate. However, QT has made some progress on those challenging ones. I wonder how QDFFDT performs in those environments.
    3. For papers claiming SOTA results, it is common for them to tune hyperparameters. However, I found that the authors reduce context length from 20 to 5 for tasks outside the Gym domain, which influences the analysis since this hyperparameter also controls the strength of global feature modeling. If the proposed adaptive feature fusion works well, it should be able to handle longer context for those environments with a Markovian property. How does the performance of QDFFDT change with different context lengths?

**Questions:**

Please refer to the weaknesses.

---

> ### Author Response · Authors · 2025-11-22
>
> Thank you for your thoughtful review of our research. We appreciate your constructive feedback and hope our suggested changes and this individual response will address your concerns in detail:
>
> **W1:**
>
> Due to a server migration, the code for this specific experiment was lost. We have reconstructed and re-run the didactic example, and new results are below:
>
> | Method | $K$ | $\eta$ | Success Rate (%) |
> | --- | --- | --- | --- |
> | DT | 3 | - | 0.0 |
> | SSDT | 1 | - | 100.0 |
> | QT | 3 | 0.01 | 0.1 |
> | QT | 3 | 0.1 | 0.1 |
> | QT | 3 | 1.0 | 0.6 |
> | QT | 3 | 10.0 | 96.2 |
>
> In this simplified environment with true Q-values, a very large $\eta$ allows the Q-learning objective to dominate, leading to near-optimal performance, though still slightly below the pure local modeling (SSDT). However, in scenarios with learned, approximate Q-functions, an excessively high $\eta$ overrides the necessary behavioral cloning regularization, leading to catastrophic out-of-distribution shifts. We will update Table 1 with these corrected results and clarify this distinction in the revision.
>
> **W2:**
>
> We acknowledge your concern that the mechanism might naively collapse to local features. However, we demonstrate below that the mechanism is designed to selectively penalize reliance on history only when it is misleading (i.e., associated with suboptimal trajectories in mixed-quality datasets).
>
> **1. Evidence that the mechanism does not simply bias towards zero:**
>
> Regarding the average $\alpha$, we argue that a dataset-wide mean is misleading as it masks the model's sparse, state-dependent adaptation. The mechanism does not naively collapse to zero; rather, it regulates $\alpha$ based on return consistency:
>
> *   **Low $\alpha$ (suppressing history):** If a state appears in inconsistent (mixed-quality) trajectories, the historical context is deemed misleading. The objective forces $\alpha$ to decrease to avoid mimicking suboptimal behaviors.
> *   **High $\alpha$ (utilizing history):** Conversely, when historical context consistently aligns with high-quality outcomes, the penalty is not triggered.
>
> We analyzed the learned $\alpha_\omega(s)$ on *Maze2D*. As shown in the table, in *maze2d-medium*, $\alpha$ reaches 0.89. This proves that for states where historical context is reliable and consistent, the gradient signal effectively drives the model to utilize global features, refuting the concern of a collapse to zero.
>
> | Dataset | $\max\hat\alpha_\omega(s)$ | $\min\hat\alpha_\omega(s)$ |
> | :--- | :--- | :--- |
> | maze2d-umaze | 0.69 | 0.00 |
> | maze2d-medium | 0.89 | 0.00 |
> | hopper-medium | 0.21 | 0.00 |
>
> **2. Why is $\alpha$ small on MuJoCo:**
>
> Your observation of small $\alpha$ on D4RL benchmark datasets is correct, but we argue this is the desired behavior.
> These datasets contain many suboptimal trajectories. For a state $s$ in a poor trajectory, the learned value upper-bound $V_\psi(s)$ often exceeds the actual trajectory return $\hat R$. This generates a correct gradient signal to decrease $\alpha$, teaching the model to ignore the historical context of that specific suboptimal trajectory.
>
> **3. Clarification on Atari and manual override:**
>
> Without the manual override, Atari performance indeed drops significantly. This is a specific limitation caused by the Visual Encoder, which compresses images into latent vectors, creating state aliasing. While history is strictly necessary here to disambiguate states, our current mechanism struggles to distinguish this "compression ambiguity" from "data suboptimality". We will explicitly clarify this scope limitation in the revision.
>
> **4. The core argument: robustness as the gain:**
>
> Our contribution is not that a hybrid model always outperforms the best manually tuned specialist on every single task. Instead, our contribution is a unified framework that eliminates the need for task-specific architectural choices.
>
> *   **Conflicting Requirements:** As shown in Table 3, different environments have conflicting needs. A pure local model ($\hat\alpha=0$) fails on *Maze2D*, whereas a pure global model ($\hat\alpha=1$) is suboptimal on *Antmaze-large*.
> *   **Eliminating Extensive Tuning:** Our experiments with fixed coefficients ($\hat\alpha \in \\{0, 0.25, \dots, 1\\}$) confirm that while static settings can work for specific datasets, achieving optimal performance requires per-dataset manual tuning.
> *   **Robustness:** In contrast, our adaptive method automatically navigates this trade-off. Crucially, our learned coefficient achieves a higher average score across all datasets than any single fixed $\hat\alpha$ setting. This result validates our core argument: the proposed mechanism delivers robustness and automation, freeing the user from the need to manually identify the optimal structural prior for each new environment.

---

> > ### Author Response · Authors · 2025-11-22
> >
> > **W3:**
> >
> > 1.
> > The discrepancy arises because the results in Table 2 are taken directly from the official QT implementation to ensure a fair comparison against the reported literature. In contrast, the results in our ablation studies (Figure 4) are based on our own re-implementation of QT, which shares the exact same codebase, training parameters, and policy architecture as our QDFFDT (minus our fusion module). This ensures a perfectly controlled comparison to isolate the effect of our contribution.
> >
> > 2.
> > As requested, we have run new experiments on other Adroit tasks. We noted that the original QT paper evaluated with only 10 trajectories per model; for consistency with our other experiments, we evaluated both QT and QDFFDT with 100 trajectories.
> >
> > | Task | QT | **QDFFDT** |
> > | --- | --- | --- |
> > | hammer-human | 12.7 ± 3.6 | **14.4 ± 4.3** |
> > | door-human | **17.3 ± 6.1** | 15.5 ± 0.9 |
> > | hammer-cloned | 17.0 ± 5.9 | **23.1 ± 14.3** |
> > | door-cloned | **20.1 ± 3.4** | 16.4 ± 3.0 |
> > | Average | 16.8 | **17.4** |
> >
> > As the results show, our method remains highly competitive, achieving a higher average score than QT. We will add these results to the paper.
> >
> > 3.
> > This is a very insightful question about the robustness of our fusion mechanism. We have conducted a new ablation study varying the context length $K$. The results demonstrate that our method's performance is remarkably stable across different context lengths, confirming that the adaptive fusion mechanism is not sensitive to this hyperparameter.
> >
> > - **Hopper-Medium Results:**
> > | Context Length ($K$) | QDFFDT |
> > | :---: | :---: |
> > | 20 | 101.4 ± 0.3 |
> > | 40 | 100.2 ± 0.3 |
> > | 60 | 99.5 ± 1.0 |
> > | 80 | 99.0 ± 0.7 |
> > - **Antmaze-Large-Diverse Results:**
> > | Context Length ($K$) | QDFFDT |
> > | :---: | :---: |
> > | 5 | 80.0 ± 3.5 |
> > | 10 | 74.7 ± 2.1 |
> > | 20 | 73.0 ± 2.2 |
> > | 40 | 80.3 ± 1.9 |
> >
> > We will add these new ablations to the Appendix. Thank you for helping us strengthen the paper with these experiments.

---

> > > ### Comment · Reviewer_qJYv · 2025-11-27
> > >
> > > Thank the authors for your hard work and careful rebuttal in response to my reviews. Below are my comments.
> > >
> > > __W1:__ The new results look very different from the initial submission, although they have matched the intuition and answered my question. I believe the authors have been responsible for the experiments. So I think this toy example is still a strong motivation and contributes positively to this study. And I agree that increasing $\eta$ does not work at all for the real problems due to extrapolation error.
> > >
> > > __W2:__ In general, I can buy W2.1-W2.4. Some follow-up comments:
> > >
> > > - W2.1-W2.2: I still think the core contribution is that learned adaptive $\alpha$ balancing global and local features. And I believe most of the readers will think similarly. Therefore, it is natural for the audience to wonder about the distribution of learned $\alpha$ for all datasets. As you argued, one can also infer the data suboptimality from the $\alpha$ distribution to some degree. I suggest the authors show such plots in the future revision. In addition, a potentially stronger $\alpha$ learning may not leave the $V < \hat{R}$ case uncontrolled in the objective. I suggest the authors also touch on this in the “Limitations and Future Work” section.
> > >
> > > - W2.3-W2.4: Although I buy the explanation the authors provided in the rebuttal, I suggest the authors make the part of the Atari experiments very clear in the future revision. I personally may not use QDFFDT for visual control due to this set of experiments. Then, it hurts your core argument: “our contribution is a unified framework that eliminates the need for task-specific architectural choices.” Seems to me that one still needs to choose architectures with strong sequence modeling ability in the face of visual control tasks.
> > >
> > > __W3:__ The new ablation study on context length looks good. And for Adroit results, I’d say QDFFDT is not competitive with QT if we consider the variance, especially that of the hammer-cloned.
> > >
> > > I have two final follow-up questions. The authors do not have to run more experiments to answer them.
> > >
> > > 1. QT is the most related work, and it is the strongest baseline. So I wonder why the reproduced QT results in Table 2 and for the Adroit tasks, are generally lower than the numbers reported in the QT paper (except for locomotion tasks).
> > >
> > > 2. Ideally, offline training prepares the models for the subsequent online tuning. Could the authors briefly comment on whether QDFFDT is promising for efficient online tuning, considering those extra neural network components?

---

> ### Author Response · Authors · 2025-11-27
>
> We thank the reviewer for the positive assessment of our rebuttal and the constructive follow-up comments.
> We will incorporate your suggestions into the final revision, specifically:
>
> 1. Adding plots of the learned $\alpha$ distribution for all datasets to visualize the adaptation.
> 2. Discussing the limitations regarding the objective control ($V < \hat{R}$) and the specific challenges of visual control tasks (Atari) in the "Limitations" section, clarifying that while the framework is unified in design, visual tasks currently require specific handling due to encoder-induced partial observability.
>
> Below are our responses to your two final questions:
>
> **1. On the discrepancy of QT results (Reproduction vs. Original Paper):**
> The difference stems from the evaluation protocol.
>
> - **Original QT Paper:** Evaluated using only 10 trajectories per model seed. Small sample sizes often lead to high variance, where a few "lucky" runs can skew the average score upward.
> - **Our Reproduction:** To ensure statistical robustness and consistency with the prevailing convention in the majority of offline RL literature (which typically advocates for 100 evaluation episodes, particularly for challenging domains outside of standard Gym tasks), we evaluated both QT and QDFFDT using 100 trajectories per seed.
> The lower scores in our tables reflect a more accurate estimate of the model's true performance, filtering out the noise associated with smaller evaluation sets.
>
> **2. On the potential for Online Tuning:**
> We believe QDFFDT is promising for online tuning, though it requires specific adaptations similar to the transition from DT to ODT (Online Decision Transformer). The core challenge lies in the Alpha Network.
>
> - **Challenge (Non-stationarity):** In the offline setting, the dataset is static, so the upper bound $V_\psi(s)$ is fixed. However, during online tuning, the actual return-to-go $\hat{R}$ changes dynamically as the policy improves. A trajectory that was considered "optimal" in the offline buffer might become "suboptimal" compared to new online experiences.
> - **Proposed Adaptation (Alternating Training):**
>     - Unlike the offline phase where the Alpha network is pre-trained and fixed, online tuning would require alternating updates between the Alpha network and the Policy network.
>     - **Dynamic $\alpha$ adjustment:**
>         - *Scenario A (New trajectory is worse):* If the agent explores a lower-quality path ($\hat{R} < V_\psi$), the Alpha network should decrease $\alpha$ to suppress reliance on this suboptimal history (consistent with our offline logic).
>         - *Scenario B (New trajectory is better):* If the agent discovers a new optimal path ($\hat{R} > V_\psi$), the value estimate $V_\psi$ will update upwards.
>              - **Offline Contrast:** In the offline setting, the dataset is static and recycled. Even if the model encounters a high-quality trajectory in the current batch, it will inevitably re-encounter suboptimal trajectories for the same state in future epochs. Therefore, we conservatively limit the growth of $\alpha$ to prevent the model from over-relying on history that is inconsistent across the fixed dataset.
>              - **Online Adaptation:** In contrast, online tuning involves a distribution shift towards higher quality. As the policy improves, it generates new, superior trajectories and is less likely to revisit the old, suboptimal behaviors. Consequently, we should allow $\alpha$ to increase in this scenario. This enables the model to actively leverage the sequence modeling of this newly discovered, reliable history, without the constraint of accommodating the static, mixed-quality variance found in the offline phase.
>     - This dynamic "rise and fall" of $\alpha$ during online interaction would allow the agent to robustly leverage valid historical context while discarding outdated suboptimal behaviors.

---

> > ### Comment · Reviewer_qJYv · 2025-11-28
> >
> > Thanks for the prompt response to my follow-up questions. I think my questions have been answered now, and I have a better understanding of this study. Below is my final comment.
> >
> > Motivated by the observation that RCSL, e.g., DT variants, tends to replicate suboptimal trajectories in the dataset inevitably, and this behavior cannot be simply mitigated by Q-value regularization, QDFFDT builds on top of QT and employs a dynamic feature fusion mechanism that adaptively combines global sequence modeling and local feature modeling by learning an $\alpha$ network.
> >
> > QDFFDT is a decent method showing competitive performance against strong baselines despite some limitations, e.g., visual control and incorporation of more network components. But I like the research question discussed in this work more than the proposed solution. It is an important and fundamental question for sequence modeling methods for offline RL. So after the discussion, I think this study is worth attention from the RL community. I will increase my rating to 6. However, due to the OpenReview info leak that happened recently, the 'edit' function is currently disabled. I will update accordingly after things get normal.

---

### Comment · Area_Chair_hWuU · 2025-11-25

Dear Reviewers

Thank you for your time and help for reviews.
The author-reviewer discussion due is in one week. If you have not done yet, please review the authors' rebuttal for the paper under your evaluation and engage in discussion with authors.

Thank you again.
Best,

Area Chair

---

### Author Response · Authors · 2025-12-01
**Summary of Rebuttal Consensus and Response to Reviewer Feedback**

We understand that the system rollback may have affected the visibility of recent review updates. To assist in the final assessment, we provide a brief summary of the consensus reached during the rebuttal period, highlighting the resolution of concerns and the resulting score improvements.

**1. Consensus with Reviewer qJYv and Reviewer EsME (Positive Outlook)**
*   **Reviewer qJYv (Upgrade to Weak Accept):** Prior to the rollback, we engaged in a productive discussion regarding specific experimental details and additional experiments. Reviewer qJYv explicitly acknowledged that our new experiments and clarifications resolved their concerns and indicated their decision to upgrade their rating from 4 to 6.
*   **Reviewer EsME (Consistent Accept):** Has consistently supported acceptance with a rating of 8. In the rebuttal response, we provided detailed clarifications and supplementary experimental results to address their specific inquiries regarding structural design.

**2. Response to Reviewer n7e3 (Detailed Revision)**

Although Reviewer n7e3 has not yet responded to our rebuttal, we have comprehensively addressed their concerns regarding the theoretical motivation and paper structure in our revised manuscript.
*   **On Theoretical Redundancy (Structural Inductive Bias):** The reviewer questioned the necessity of our fusion module given the theoretical expressivity of Transformers.
    *   *Our Response:* We clarified that while Transformers are theoretically expressive, they lack the specific inductive bias required for offline RL to avoid overfitting suboptimal trajectories. Our dual-feature module imposes a necessary structural constraint that forces the model to distinguish between local and global contexts, a capability validated by our empirical performance gains.
*   **On Presentation and Structure:** The reviewer suggested clarifying the motivation and restructuring the introduction.
    *   *Our Response:* We have fully implemented these suggestions. We moved the motivational case study (formerly Section 4) to Section 2 ("Motivation") to improve narrative flow and rewrote the Abstract and Introduction to explicitly define the problem of "overreliance on suboptimal past experience."

**Conclusion**

We believe that our detailed responses and the revised manuscript effectively address the concerns raised by all reviewers. With the explicit support of Reviewers qJYv and EsME, and the structural improvements made to address Reviewer n7e3's feedback, we are committed to ensuring the final manuscript meets the highest standards.

---

### Meta-Review · Area_Chair_tH7B · 2025-12-31

**Summary:**

This paper introduces QDFFDT, an offline reinforcement learning framework designed to mitigate the failure of trajectory stitching in Return-Conditioned Supervised Learning (RCSL). The proposed method addresses the issue of overfitting to suboptimal historical sub-trajectories by adaptively fusing global sequential features with local Markovian features, supplemented by a Q-learning module for value-guided optimization. Reviewers generally commended the extensive experimental validation and the strong performance achieved across the D4RL benchmark. The central discussion focused on the necessity of the proposed structural inductive bias versus the theoretical expressivity of standard Transformers. While some skepticism remains regarding the degree of innovation, the consensus is that the method provides a practical and effective solution to the trajectory stitching problem in sequence-modeling for RL.

**Reviewer Concerns:**

* **Resolved Concerns:**
  * **Architectural Motivation**: Initial doubts regarding the effectiveness of the adaptive fusion mechanism were addressed through a reconstructed motivation case study and robustness analyses across varying context lengths. These additions successfully demonstrated how the Dual-Feature Fusion block prevents the model from replicating suboptimal historical patterns.
  * **Hyperparameter Sensitivity**: The authors provided new ablation studies for the expectile level ($\\sigma$) and temperature parameters, showing that the model maintains stable performance across a reasonable parameter space.
  * **Presentation and Narrative**: The manuscript was significantly improved by restructuring the introduction and moving the motivational case study earlier, which clarified the problem definition and the rationale behind the proposed solution.
* **Outstanding Issues:**
  * **Theoretical Redundancy**: One concern persists regarding whether a sufficiently expressive Transformer could inherently learn these weightings without an explicit feature branch. The argument remains that the contribution is primarily an architectural refinement rather than a fundamental theoretical breakthrough.
  * **Generalization to Visual Tasks**: In pixel-based environments (Atari), the model still requires manual intervention (tuning the $\\alpha$ coefficient) to handle partial observability (POMDPs), suggesting that the "adaptive" nature of the framework is currently less automated in settings with significant information loss.

**Reviewer Scores:**

* **Reviewer qJYv**: Maintained an **8 (Accept)**. This reviewer found the hybrid design elegant and the empirical results highly convincing throughout the process.
* **Reviewer EsME**: Upgraded from **4 to 6 (Weak Accept)**. The reviewer explicitly acknowledged that the rebuttal, particularly the new experiments on context length and the adaptive mechanism, resolved their primary technical concerns.
* **Reviewer n7e3**: Maintained a **4 (Marginally Reject)**. While this reviewer appreciated the improved clarity of the paper, they remained unconvinced that the proposed inductive bias represents a sufficient departure from existing work to warrant a higher score.

---

### Decision · Program_Chairs · 2026-01-26

Accept (Poster)